# Predicting response to immunotherapy in gastric cancer via multi-dimensional analyses of the tumour immune microenvironment

Yang Chen [1,8], Keren Jia [1,8], Yu Sun [2,8], Cheng Zhang [1,8], Yilin Li [1], Li Zhang [3,4], Zifan Chen [3,4], Jiangdong Zhang [5], Yajie Hu [4], Jiajia Yuan [1], Xingwang Zhao [1], Yanyan Li [1], Jifang Gong [1], Bin Dong [6,7], Xiaotian Zhang [1], Jian Li [1] & Lin Shen [1] ✉

A single biomarker is not adequate to identify patients with gastric cancer (GC) who have the potential to benefit from anti-PD-1/PD-L1 therapy, presumably owing to the complexity of the tumour microenvironment. The predictive value of tumour-infiltrating immune cells (TIICs) has not been definitively established with regard to their density and spatial organisation. Here, multiplex immunohistochemistry is used to quantify in situ biomarkers at subcellular resolution in 80 patients with GC. To predict the response to immunotherapy, we establish a multi-dimensional TIIC signature by considering the density of $CD4^+FoxP3^-PD-L1^+$, $CD8^+PD-1^-LAG3^-$, and $CD68^+STING^+$ cells and the spatial organisation of $CD8^+PD-1^+LAG3^-$ T cells. The TIIC signature enables prediction of the response of patients with GC to anti-PD-1/PD-L1 immunotherapy and patient survival. Our findings demonstrate that a multi-dimensional TIIC signature may be relevant for the selection of patients who could benefit the most from anti-PD-1/PD-L1 immunotherapy.

Gastric cancer (GC) is the fifth most common cancer in the world and the second leading cause of cancer-related deaths[1]; more than 47% of all cases occur in China. The advent of immune checkpoint inhibitors (ICIs), targeting programmed cell death protein 1 (PD-1) and programmed death-ligand 1 (PD-L1), has revolutionised cancer therapy, providing robust and durable responses in GC. Clinical trials with pembrolizumab or nivolumab monotherapy have demonstrated a wide range of response rates in advanced GC (10–26%), which has no selective biomarker[2–4]. Hence, to enhance the efficacy of anti-PD-1/PD-L1 therapy in GC, there is an urgent need to identify the patients who are most likely to benefit from immunotherapy.

Many biomarkers, including tumour mutation burden (TMB), PD-L1 expression, microsatellite instability (MSI) and Epstein–Barr virus (EBV) infection status, have been proposed to identify susceptibility to PD-1/PD-L1 inhibitors[5,6]. However, the results of several clinical trials using these biomarkers at an individual level are not consistent; some are even contradictory[7–9]. Therefore, to date, no single biomarker is available for adequate patient stratification (not only in GC), presumably due to the complexity of the immune response to cancer.

Tumour immune cells are heterogeneous, show functional and phenotypic plasticity and may exert both pro-tumourigenic and anti-tumourigenic effects[10]. Interestingly, the distribution of different subsets of immune cells and their precise location in relation to cancer

[1]Department of Gastrointestinal Oncology, Key Laboratory of Carcinogenesis and Translational Research (Ministry of Education), Peking University Cancer Hospital and Institute, Beijing, China. [2]Department of Pathology, Key Laboratory of Carcinogenesis and Translational Research (Ministry of Education), Peking University Cancer Hospital and Institute, Beijing, China. [3]Center for Data Science, Peking University, Beijing, China. [4]National Biomedical Imaging Center, Peking University, Beijing, China. [5]School of Software and Microelectronics, Peking University, Beijing, China. [6]Beijing International Center for Mathematical Research (BICMR), Peking University, Beijing, China. [7]Center for Machine Learning Research, Peking University, Beijing, China. [8]These authors contributed equally: Yang Chen, Keren Jia, Yu Sun, Cheng Zhang. ✉e-mail: shenlin@bjmu.edu.cn

cells have been proposed as valuable indicators for the prediction of tumour behaviour[11,12]. In fact, the components of the tumour microenvironment that may influence the therapeutic response, including T cells, B cells, neutrophils and macrophages, are increasingly being considered in the field[13–16]. For instance, previous studies have shown that knocking down cyclic GMP-AMP synthase-stimulator of interferon genes (STING) promotes the polarisation of tumour-associated macrophage (TAM) into the pro-inflammatory subtype and induces the apoptosis of GC cells, highlighting the negative function of STING in TAMs[17]. Therefore, analysing the spatial relationships of individual cellular and acellular components may advance the understanding of GC biology and facilitate the development of improved tumour immune biomarkers. The accurate identification of specific tumour immune cell subsets requires a combination of multiple markers. Importantly, recent developments in multiplex immunohistochemistry (m-IHC) have enabled the simultaneous detection of multiple antigens in situ at a single-cell resolution[18]. However, these methods have not been used to analyse immune cells in GC in the context of immunotherapy; most analyses have been conducted using traditional IHC.

In this work, we use m-IHC combined with digital image analysis and machine learning to identify the immune cell features of GC clinical specimens. We characterise the density and spatial patterns of tumour-infiltrating immune cells (TIICs), their variation depending on the GC molecular feature, as well as their prognostic significance. These data will help to evaluate the density and spatial patterns of TIICs in the context of anti-PD-1/PD-L1 treatment for a better understanding of the determinants of response to immunotherapy in GC.

## Results

### Clinico-pathological features of the GC patients

Eighty patients were enrolled in this study between July 2014 and December 2019 (Table 1). The median age of the patients was 60 years (range, 54–66 years), and most patients were men (76.3%). Among the 60 patients subjected to immunotherapy, 21 were treated with standard-of-care anti-PD-1/PD-L1 antibodies and 39 were treated as part of clinical trials (NCT03472365, NCT03713905). Archived pretreatment samples were available from all patients. Ten (12.5%) patients were EBV(+) and 11 (13.75%) had confirmed deficient DNA mismatch repair (dMMR) GC.

### TIIC distribution: analysis overview

To investigate the landscape of TIICs within the GC specimens, we quantified the density and spatial location of immune cells in 80 full-face formalin-fixed paraffin-embedded (FFPE) samples via m-IHC staining; the multiplex determination of the sub-cellular expression of 16 proteins was performed (Fig. 1a). First, haematoxylin and eosin (H&E)-stained tissue sections were reviewed by two pathologists (S.Y. and H.Y.J.) to identify tumour core (TC), invasion margin (IM), and peritumoural normal (N) areas, which we refer to as regions of interest (ROIs) (Fig. 1b). The m-IHC panels analysed are depicted in Fig. 1c−f. A total of 6488 high-power fields (TC: 4477, IM: 993, N: 1018) were imaged for all patients. A supervised image analysis system (inForm) was used to classify each image into tumour nests and stromal areas based on machine learning (Fig. 1g). Cell segmentation showed nuclear, cytoplasmic and membranous outlines. Cell phenotyping data were obtained based on the positivity and relative intensity of all markers in one panel. The cell density, calculated for "all" regions (tumour + stroma), was measured separately in the tumour and stroma. Thereafter, TIICs were analysed at the single-cell level and 26 major populations were characterised (Supplementary Fig. 1a).

### TIICs are differentially distributed across distinct ROIs

To examine the distribution of TIICs within the tumour microenvironment, we analysed their spatial density in the TC, IM, and N areas. The enriched co-occurrence of immune populations defines a

**Table 1 | Baseline characteristics of gastric cancer patients**

| Characteristic[*] | Total N = 80 | Immunotherapy N = 60 |
|---|---|---|
| **Age** | | |
| Median, IQR | 60 (54–66) | 59.5 (50.5–66) |
| **Sex (Male/Female)** | | |
| Male | 61 (76.3%) | 46 (76.7%) |
| Female | 19 (23.7%) | 14 (23.3%) |
| **ECOG PS** | | |
| 0 | 49 (61.3%) | 38 (63.3%) |
| 1 | 31 (38.7%) | 22 (36.7%) |
| **Location** | | |
| GEJ | 24 (30.0%) | 19 (31.7%) |
| Non-GEJ | 56 (70.0%) | 41 (68.3%) |
| **Differentiation** | | |
| Moderate | 23 (28.8%) | 17 (28.3%) |
| Moderate-poor | 22 (27.5%) | 19 (31.7%) |
| Poor | 35 (43.7%) | 24 (40.0%) |
| **Lauren classification** | | |
| Intestinal type | 38 (47.5%) | 28 (46.7%) |
| Diffused type | 18 (22.5%) | 13 (21.7%) |
| Mixed type | 24 (30.0%) | 19 (31.7%) |
| **Stage** | | |
| I | 3 (3.8%) | 3 (5.0%) |
| II | 9 (11.3%) | 8 (13.3%) |
| III | 29 (36.2%) | 18 (30.0%) |
| IV | 39 (48.7%) | 31 (51.7%) |
| **HER2 expression** | | |
| Positive | 22 (27.5%) | 15 (25%) |
| Negative | 58 (72.5%) | 45 (75%) |
| **PD-L1 expression (CPS)** | | |
| ≥10 | 36 (45.0%) | 27 (45%) |
| 5-10 | 10 (12.5%) | 8 (13.3%) |
| 1-5 | 17 (21.25%) | 13 (21.7%) |
| <1 | 17 (21.25%) | 12 (20%) |
| **MMR status** | | |
| pMMR | 69 (86.25%) | 52 (86.7%) |
| dMMR | 11 (13.75%) | 8 (13.3%) |
| **EBV status** | | |
| Positive | 10 (12.5%) | 9 (15.0%) |
| Negative | 70 (87.5%) | 51 (85.0%) |

*dMMR* deficient mismatch repair, *pMMR* proficient mismatch repair, *CPS* combined positive score.

[*]Percentage indicates the proportion of patients with a specific clinical, pathologic, or molecular characteristic among all patients.

structured immune environment (Supplementary Fig. 1a). A significant increase in the overall density of CD68[+] cells was observed within the TC compared with that in the adjacent normal tissues; an opposite trend was observed for CD8[+] and CD20[+] cells (Fig. 2a). Next, for a higher degree of detail, the distribution of each TIIC was explored. CD8[+], CD8[+]PD-1[−]LAG-3[−], CD20[+] and CD68[+]CD163[+]HLA-DR[−] cells accumulated at the IM and decreased toward the TC. In contrast, CD8[+]PD-1[+]TIM-3[+], CD8[+]PD-1[−]TIM3[+], CD8[+]PD-1[+]LAG-3[+]TIM-3[+], CD8[+]PD-1[+]LAG-3[−]TIM-3[+], CD4[+]FoxP3[+]CTLA-4[+], CD4[+]FoxP3[−]CTLA-4[+], CD68[+], CD68[+]HLA-DR[+]CD163[−] cells accumulated at the TC and decreased toward the IM. Interestingly, a higher density of CD4[+]FoxP3[+] and CD4[+]FoxP3[+]PD-L1[+] cells was found within the TC than in normal tissues (Fig. 2b, Supplementary Fig. 1b), highlighting the heterogeneous distribution of TIICs in GC.

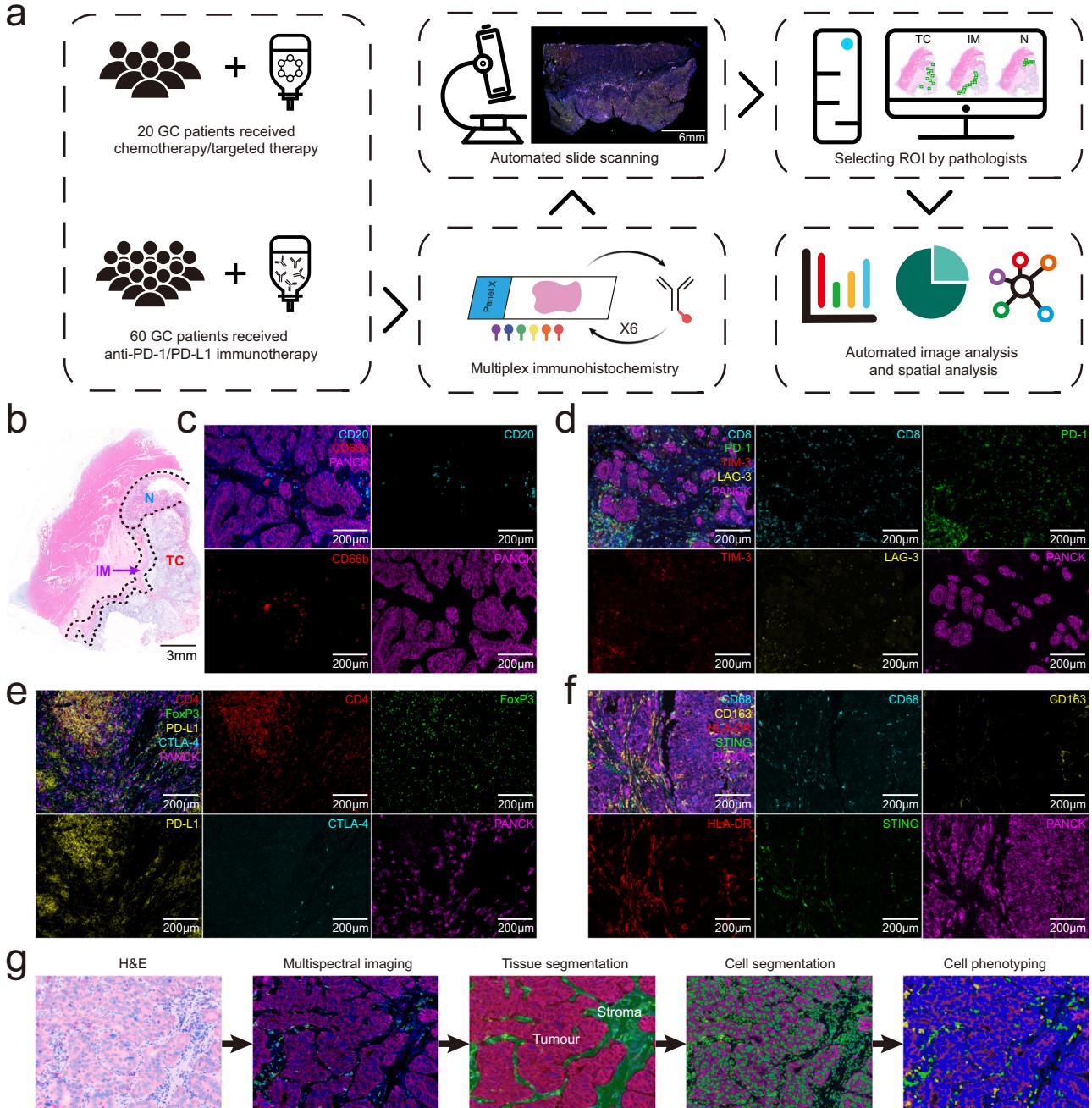

**Fig. 1 | Identification and characterisation of tumour-infiltrating immune cells in gastric cancer tissues. a** Schematic representation of the experimental design and analytical methods used in this study. **b** Selection of the regions of interest (ROIs) in representative images of haematoxylin and eosin (H&E)-stained formalin-fixed paraffin-embedded tissues. TC, tumour core; IM, invasion margin; N, normal tissue. Scale bar: 3 mm. **c–f** Representative composite and single-stained images of the multiplex immunohistochemistry panels used. Scale bar: 200 μm. **g** Overview of the automated image analysis pipeline.

Additionally, the localisation of TIICs with respect to the tumour nest and stroma areas (defined in Fig. 1g) was further examined. CD8[+], CD4[+] and CD20[+] cells were located primarily in the stroma and were less prevalent in the tumour nest. In contrast, CD66b[+] cells were more prevalent in the tumour nest than in the stroma (Supplementary Fig. 2a).

**The infiltration profile of TIICs is different in distinct molecular GC subtypes**

To evaluate the tumour immune microenvironment in GC, we compared the density of TIICs in the context of distinct clinico-pathological factors (Fig. 2c, Supplementary Fig. 3a–e). Generally, there were few significant differences between Lauren classification,

tumour differentiation and tumour location (oesophagogastric junction or not) with respect to densities of TIICs (Supplementary Tables 1–5, Supplementary Fig. 4a). Additionally, there were few differences in the density of TIICs between HER2-positive and -negative GC (Fig. 2c). Overall, the density of total CD8[+], CD4[+] and CD68[+] cells was associated with the disease stage. Additionally, advanced-stage GC (III-IV) samples showed a higher density of exhausted CD8[+] T cells, CD4[+]FoxP3[−] cells and so on.

Furthermore, we analysed the density of TIICs in GC of different molecular subtypes (Supplementary Tables 6–8). Interestingly, EBV-positive tumours showed higher densities of CD8[+]PD-1[−]LAG-3[−] T cells than EBV-negative ones. EBV (+) GCs were characterised by abundant

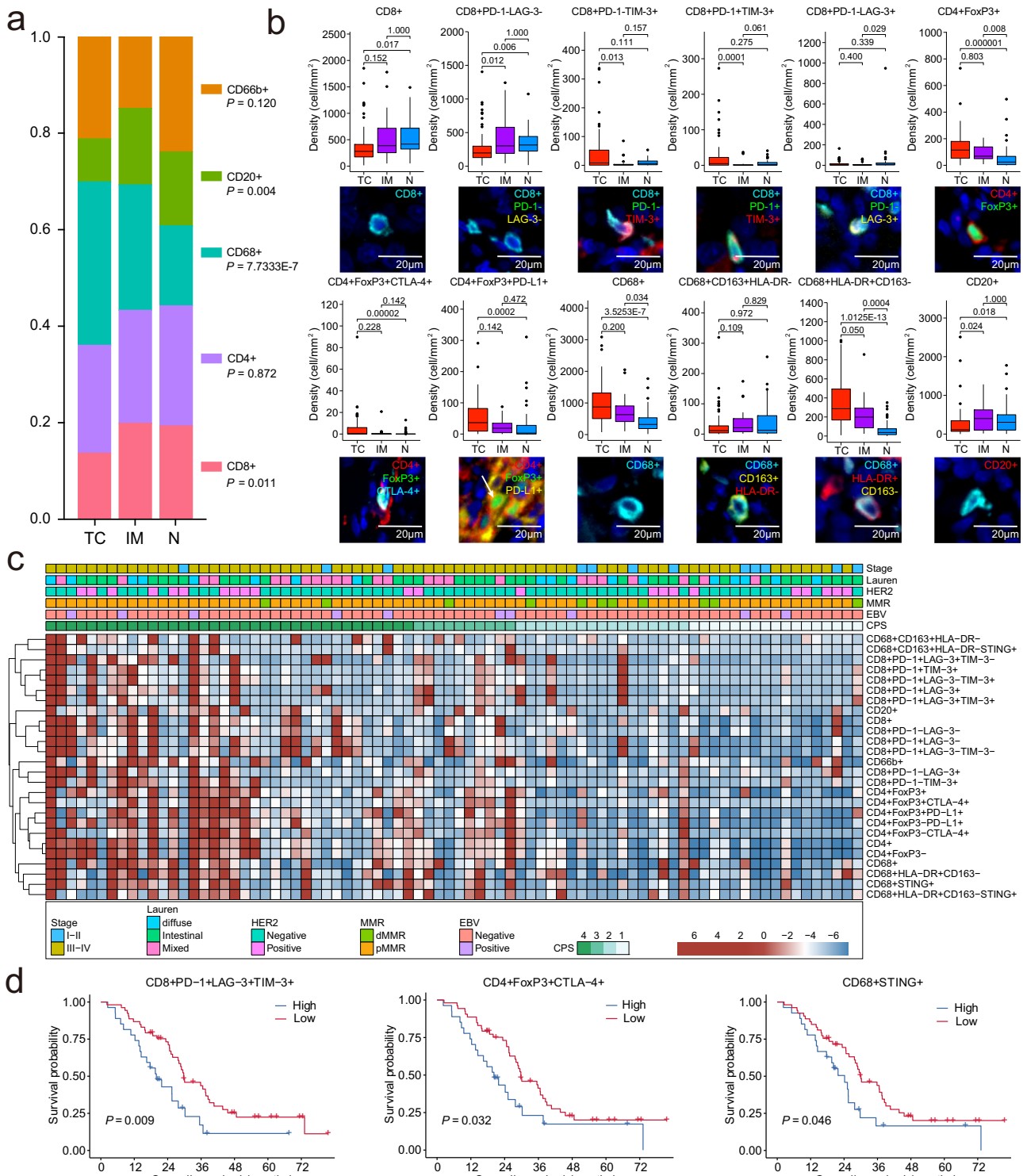

**Fig. 2 | Automated image analysis highlights the ordered immune composition in gastric cancer. a** Constitution of the main tumour-infiltrating immune cell (TIIC) populations. Kruskal–Wallis test with the Dunn's multiple comparison test. **b** Density of TIICs across the regions of interest (*n* = 80). TC, tumour core; IM, invasion margin; N, normal tissue. Immunofluorescence staining images refer to the co-expression of the corresponding markers and DAPI (nuclei). Scale bar: 20 μm. Box and whiskers represent mean ± 10–90 percentile. Kruskal–Wallis test with Dunn's multiple comparison test. **c** TIIC density grouped by subtypes. **d** Overall survival of 80 patients based on the density of TIICs. The individual TIICs were divided into high (>two-thirds of the patients; blue line) or low density (≤two-thirds of patients; red line). Log-rank (Mantel–Cox) test was used. A two-sided *P* < 0.05 was considered statistically significant.

immune cell infiltration; however, not all EBV (+) patients responded to immunotherapy, indicating that specific immune cell infiltration is needed. Proficient MMR (pMMR) tumours showed a significantly higher abundance of total CD4⁺, CD68⁺, CD20⁺ and CD66b⁺ cells than

dMMR tumours. Higher CD68⁺ and CD66b⁺ cells (neutrophils) are known to contribute to resistance to PD-1/PD-L1 treatment in several cancers[13,19]. We classified patients into four combined positive score (CPS) groups: CPS < 1, 1 ≤ CPS < 5, 5 ≤ CPS < 10 and CPS ≥ 10.

Remarkably, the abundance of TIICs, including CD8$^+$, CD4$^+$, CD68$^+$, CD20$^+$ and CD66b$^+$ cells, significantly increased with the increase in CPS, indicating a 'hotter' tumour immune environment. However, the comparison between CPS 5-10 and CPS ≥ 10 did not show a significant difference, providing evidence for the cut-off selection in clinical trials of anti-PD-1/PD-L1-based therapies. Altogether, as shown in Fig. 2c, our results suggest that the infiltration pattern of immune cells depends on, but is not restricted to, GC molecular subtypes.

### Survival analyses according to the infiltration density of TIICs

Next, we sought to understand whether the number of TIICs is correlated with patient survival. We found that higher levels of tumour-infiltrating T cell subsets, including CD8$^+$PD-1$^+$LAG-3$^+$TIM-3$^+$, CD4$^+$FoxP3$^+$CTLA-4$^+$ T and CD68$^+$STING$^+$ cells, were associated with inferior overall survival (OS) in 80 patients (Fig. 2d, Supplementary Fig. 4b). CD8$^+$PD-1$^+$LAG-3$^+$TIM-3$^+$ cells [high vs. low, hazard ratio (HR) 1.98, 95% confidence interval (CI; 1.12–3.50)] and CD68$^+$STING$^+$ cells [high vs. low, HR 1.83, 95%CI (1.01–3.33)] were significantly associated with OS, as revealed by multivariate Cox analysis (Supplementary Table 9). Collectively, these data highlight the clinical relevance of tumour-infiltrating T cells in the survival of GC patients.

Additionally, we analysed the prognostic value of the density of TIICs in the context of tumour and stromal cells. The data showed a similar trend for CD4$^+$FoxP3$^-$CTLA-4$^+$ T and CD4$^+$FoxP3$^+$CTLA-4$^+$ T cells in both contexts. However, higher infiltration of CD8$^+$PD-1$^+$LAG-3$^+$TIM-3$^+$ T cells and CD68$^+$ macrophages was associated with poorer OS with respect to tumour nests. In addition, higher infiltration of CD8$^+$PD-1$^+$TIM-3$^+$ T cells, CD66b$^+$ neutrophils and CD68$^+$STING$^+$ macrophages was related to a shorter OS with respect to the stroma (Supplementary Fig. 2b). Therefore, these results highlight the value of studying immune cell density in defined tissue regions.

### Spatial analysis of GC shows a hierarchical organisation of tumour and immune cells

Given our ability to precisely define the positions of individual tumour cells and TIICs, we next sought to evaluate the clinical significance of the proximity between them. The observation that certain TIICs, including CD68$^+$ cells, were enriched in the tumour region suggested that the proximity of TIICs to tumour cells might influence their phenotype. To further study these localisation patterns, a bioinformatics tool (pdist; see Methods) that determines the nucleus-to-nucleus distances between any two cell types was used. To incorporate both cell proximity and quantity, an 'effective score' parameter was established: the proportion of TIICs near tumour cells (within the defined distance criteria introduced; Fig. 3a). In other words, this score was calculated by the number of paired immune cells and tumour cells divided by the total number of immune cells across the whole slides to maintain the spatial variation to a large extent. Therefore, using this formula, a higher effective score indicates that within a certain distance, there is a higher density of tumour cells around the immune cells. Importantly, across the three distances considered (0–10/0–20/0–30 μm), CD8$^+$PD-1$^+$LAG-3$^+$ T cells and CD66b$^+$ neutrophils were the ones with higher effective scores (Fig. 3b).

We also calculated the distance between each TIIC and the closest tumour cell. Neutrophils, B cells and macrophages were located closer to tumour cells. We then analysed the distances between TIICs and tumour cells according to the PD-L1 CPS. In general, TIICs were located closer to tumour cells in patients with CPS ≥ 10 (compared with the picture with respect to all other groups; Supplementary Fig. 6a).

Interestingly, the effective scores also differed between different GC molecular subtypes, including those depending on the EBV, PD-L1 CPS, MMR and HER2 status (Supplementary Figs. 5a–e, 6b; Supplementary Tables 10–17). For instance, a significantly higher effective score of exhausted T cells (CD8$^+$PD-1$^+$LAG-3$^+$TIM-3$^-$, CD8$^+$PD-1$^-$TIM-3$^+$), M1 (CD68$^+$CD163$^+$HLA-DR$^-$) and M2 (CD68$^+$HLA-DR$^+$CD163$^-$)

macrophages within a 20 μm radius was observed in HER2-negative GC compared with that in HER2-positive GC (Fig. 3c, Supplementary Fig. 5b).

### Cancer cell-adjacent TIICs are correlated with patient survival

The combination of multiplexed imaging and machine learning implied that the density of TIICs within GC is linked to patient survival. For further detail, the effective density (the absolute number of TIICs near tumour cells within a 20 μm radius) was used as an additional measurement. This radius was pre-selected to identify immune cell populations most likely capable of effective, direct, cell-to-cell interactions with tumour cells, consistent with prior studies in multiple gastrointestinal tumour types[11,20,21]. Curiously, we found that patients with higher effective densities (radius of 0–20 μm) of CD68$^+$STING$^+$ macrophages, CD68$^+$HLA-DR$^+$CD163$^-$ STING$^+$ macrophages and neutrophils showed significantly shorter OS than those with lower effective densities (Fig. 3d). Importantly, the prognostic value was still significant after adjustment using the multivariate Cox model (Supplementary Table 18). Other immune cell phenotypes were not associated with OS (Supplementary Figs. 6c and 7c). These results indicate that the influence of TIICs on patient survival is dependent not only on the number of TIICs but also on their proximity to tumour cells. Overall, our data highlight that both the location and density of TIICs should be taken into consideration for prognosis predictions.

### Multi-dimensional TIIC signature predicts response to immunotherapy

Human tumours contain exhausted T cells expressing multiple immune checkpoints; it has been proposed that these cells mediate resistance to PD-1 blockade. Thus, next, we investigated whether the density of TIICs and respective effective scores were associated with the clinical outcomes of anti-PD-1/PD-L1 immunotherapy. All 60 patients who received immunotherapy were assigned to the training (n = 44, generated retrospectively from 15/11/2016 to 17/7/2019) and validation (n = 16, generated prospectively from 29/7/2019 to 19/12/2019) cohorts. Importantly, we ensured that the clinical characteristics of the training and validation cohorts were balanced (Table 2). We used logistic regression analysis to assess the association between TIICs and the objective response rate (ORR) in the training cohort. Importantly, we found that the density of CD4$^+$FoxP3$^-$PD-L1$^+$ T cells and the effective score of CD8$^+$PD-1$^+$LAG-3$^-$ T cells were closely associated with a positive response to anti-PD-1/PD-L1 therapy; conversely, CD8$^+$PD-1$^-$LAG-3$^-$ T cells and CD68$^+$STING$^+$ macrophages were closely associated with a negative response to anti-PD-1/PD-L1 therapy (Supplementary Table 19).

The density of CD4$^+$FoxP3$^-$PD-L1$^+$ T cells, CD8$^+$PD-1$^-$LAG3$^-$ T cells and CD68$^+$STING$^+$ macrophages, and the effective score of CD8$^+$PD-1$^+$LAG3$^-$ T cells were used to define a TIIC signature (Fig. 4a), with the potential to improve the ability of identifying responders to anti-PD-1/PD-L1 immunotherapy. We used four types of machine learning models and calculated the area under the curve (AUC) of the training and validation cohorts, including extra tree classifier (ETC), AdaBoost classifier (ABC), gradient boosting classifier (GBC) and multi-layer perceptron (MLP) models. In the validation cohort, the average AUCs of the four algorithms were 0.80, 0.85, 0.77 and 0.75, respectively (Fig. 4b, c, Supplementary Table 20). The corresponding 95% CIs were narrow, suggesting that the TIIC signature can indeed be used to predict the response to immunotherapy (Supplementary Table 20). Importantly, the four algorithms showed a similar performance before and after adjusting for the hyper-parameters, indicating the strength of the predictive value of the TIIC signature itself (Supplementary Fig. 7a). Furthermore, we explored the predictive power of the TIIC score combined with CPS, EBV status and MMR status. The combined TIIC signature had a better AUC in the ETC, GBC and ABC models, but not in the MLP model (Supplementary Table 21).

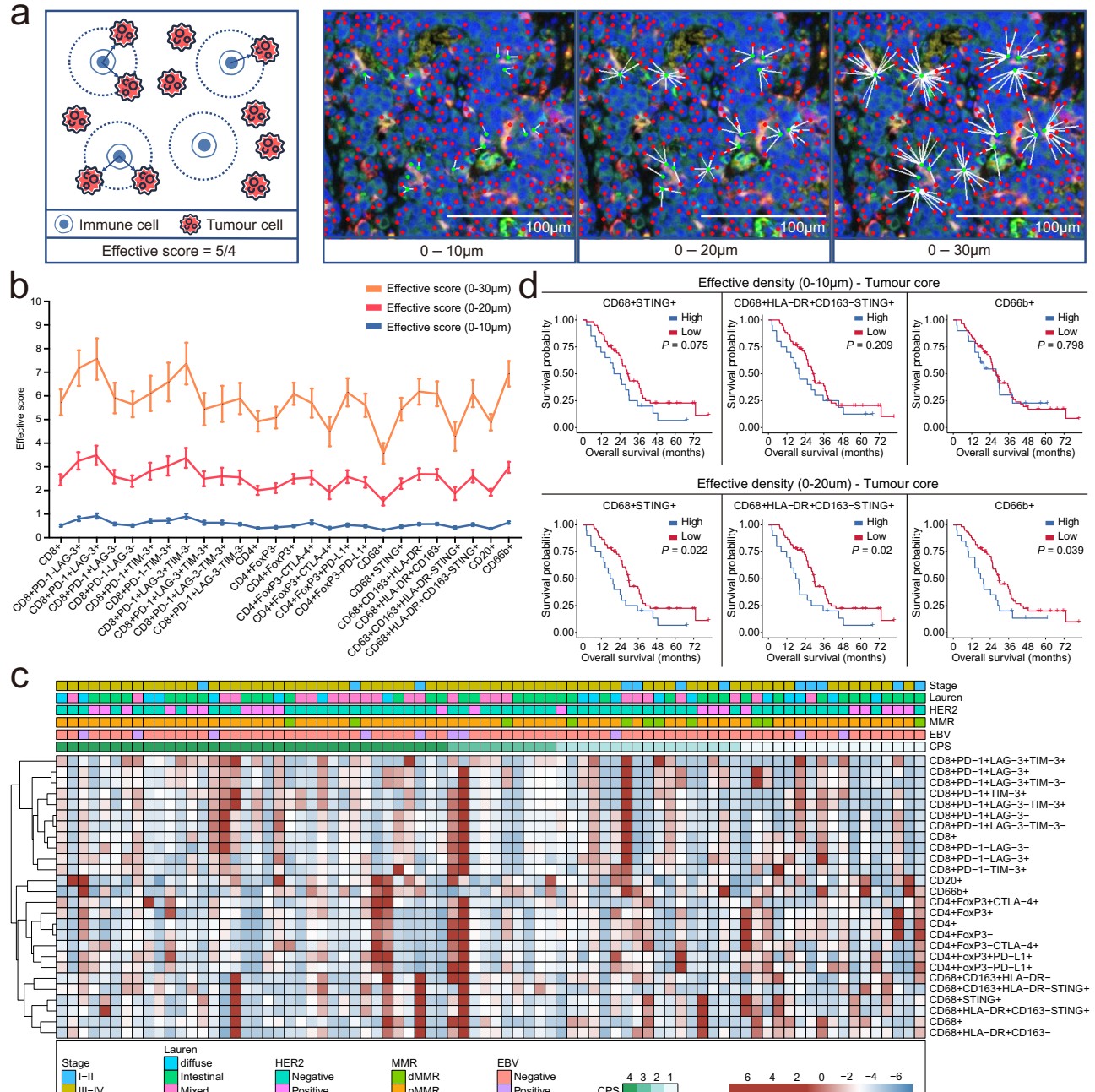

**Fig. 3 | Spatial analysis of gastric cancer shows a hierarchy of organisation of tumour and immune cells. a** Illustration of the distance analysis involving immune and tumour cells. Red dots: tumour cells; green dots: immune cells. The white translucent circle represents the radius. Effective score = number of paired immune cells and tumour cells/number of immune cells. Scale bar: 100 μm. **b** The distribution of the effective score of tumour-infiltrating immune cell (TIIC) populations in the tumour core in 10-, 20- and 30 μm increments (n = 80). Error bars represent mean ± SEM. **c** Effective score of TIICs in patients grouped by gastric cancer subtypes. EBV, Epstein–Barr virus status; MMR, DNA mismatch repair; CPS, combined positive score. **d** Overall survival of the 80 patients based on the effective densities (0–10 μm and 0–20 μm) of TIICs. The individual immune infiltrate values were divided into high (> two-thirds of the patients in the cohort; blue line) or low density (≤ two-thirds of patients in the cohort; red line). Statistical relevance was defined using the log-rank (Mantel–Cox) test. A two-sided $P < 0.05$ was considered statistically significant.

We quantified the contribution of each marker in the prediction models through feature importance using the scikit-learn package (Supplementary Tables 22, 23). We outputted the feature importance and the average value of each parameter to present its contribution. As shown in Fig. 5a, the effective score of CD8+PD-1+LAG-3− cells had higher feature importance than the density of CD68+STING+, CD4+FoxP3−PD-L1+, or CD8+PD-1−LAG-3− cells in ETC, GBC and ABC machine learning models. As presented in Fig. 5b, the effective score of CD8+PD-1+LAG-3− cells had higher feature importance than that of the other three immune cell types, EBV, MMR and PD-L1 CPS. Thus, the dominant predictive marker is the spatial organisation for response to immunotherapy.

We also evaluated the predictive values of other candidate biomarkers. AUCs of 0.58 and 0.76 in the training and validation cohorts, respectively, were defined for PD-L1 CPS (Supplementary Fig. 7b). We analysed the treatment response based on EBV status, MMR status and HER2 expression in univariate and multivariable logistic regression models (Supplementary Table 24). EBV-positive status and dMMR

**Table 2 | Baseline characteristics of 59 gastric cancer patients who received immunotherapy in the training cohort and validation cohort**

| Characteristic[*] | All N = 59 | Training cohort N = 44 | Validation cohort N = 15 | P value[§] |
|---|---|---|---|---|
| **Age** | | | | |
| Median, IQR | 60 (50–66) | 59.5 (52.3–65) | 63 (46–70) | 0.43 |
| **Sex (Male/Female)** | | | | |
| Male | 45 (76.3%) | 35 (79.6%) | 10 (66.7%) | 0.31 |
| Female | 14 (23.7%) | 9 (20.4%) | 5 (33.3%) | |
| **ECOG PS** | | | | |
| 0 | 37 (62.7%) | 27 (61.4%) | 10 (66.7%) | 0.71 |
| 1 | 22 (37.3%) | 17 (38.6%) | 5 (33.3%) | |
| **Location** | | | | |
| GEJ | 18 (30.5%) | 12 (27.3%) | 6 (40.0%) | 0.36 |
| Non-GEJ | 41 (69.5%) | 32 (72.7%) | 9 (60.0%) | |
| **Differentiation** | | | | |
| Moderate | 17 (28.8%) | 9 (20.5%) | 8 (53.3%) | 0.019 |
| Moderate-poor | 18 (30.5%) | 17 (38.6%) | 1 (6.7%) | |
| Poor | 24 (40.7%) | 18 (40.9%) | 6 (40.0%) | |
| **Lauren classification** | | | | |
| Intestinal type | 27 (45.8%) | 15 (34.1%) | 12 (80.0%) | 0.003 |
| Diffused type | 13 (22.0%) | 10 (22.7%) | 3 (20.0%) | |
| Mixed type | 19 (32.2%) | 19 (43.2%) | 0 (0.0%) | |
| **Stage** | | | | |
| I | 3 (5.1%) | 3 (6.8%) | 0 (0.0%) | 0.71 |
| II | 7 (11.9%) | 5 (11.4%) | 2 (13.3%) | |
| III | 18 (30.5%) | 14 (31.8%) | 4 (26.7%) | |
| IV | 31 (52.5%) | 22 (50.0%) | 9 (60.0%) | |
| **HER2 expression** | | | | |
| Positive | 14 (23.7%) | 41 (93.2%) | 4 (26.7) | 1.70E–7 |
| Negative | 45 (76.3%) | 3 (6.8%) | 11 (73.3%) | |
| **PD-L1 expression (CPS)** | | | | |
| ≥10 | 26 (44.1%) | 19 (43.2%) | 7 (46.7%) | 0.064 |
| 5–10 | 8 (13.6%) | 8 (18.2%) | 0 (0.0%) | |
| 1-5 | 13 (22.0%) | 11 (25.0%) | 2 (13.3%) | |
| <1 | 12 (20.3%) | 6 (13.6%) | 6 (40.0%) | |
| **MMR status** | | | | |
| pMMR | 51 (86.4%) | 37 (84.1%) | 14 (93.3%) | 0.37 |
| dMMR | 8 (13.6%) | 7 (15.9%) | 1 (6.7%) | |
| **EBV status** | | | | |
| Positive | 9 (15.3%) | 8 (18.2%) | 1 (6.7%) | 0.28 |
| Negative | 50 (84.7%) | 36 (81.8%) | 14 (93.3%) | |
| **Line of therapy** | | | | |
| 1 | 33 (55.9%) | 24 (54.6%) | 9 (60.0%) | 0.45 |
| 2 | 16 (27.1%) | 11 (25%) | 5 (33.3%) | |
| ≥3 | 10 (17.0%) | 9 (20.5%) | 1 (6.7%) | |
| **Type of anti-PD-1/PD-L1 therapy** | | | | |
| Monotherapy | 19 (32.2%) | 18 (40.9%) | 1 (6.8%) | 2.67E–7 |
| **Combination therapy** | | | | |
| chemotherapy | 18 (30.5%) | 17 (38.6%) | 1 (6.7%) | |
| Anti-VEGF | 4 (6.8%) | 4 (9.1%) | 0 (0.0%) | |
| Anti-CTLA-4 | 2 (3.4%) | 2 (4.6%) | 0 (0.0%) | |
| Anti-HER2 | 16 (27.1%) | 3 (6.8%) | 13 (86.7%) | |
| **ORR** | | | | |
| CR/PR | 19 (32.2%) | 14 (31.8%) | 5 (33.3%) | 0.91 |
| SD/PD | 40 (67.8%) | 30 (68.2%) | 10 (66.7%) | |

*dMMR* deficient mismatch repair, *pMMR* proficient mismatch repair, *CPS* combined positive score.

[*]Percentage indicates the proportion of patients with a specific clinical, pathologic, or molecular characteristic among all patients.

[§]To compare characteristics between subgroups, we used the $\chi^2$ test for categorical variables and Mann–Whitney U test for non-normally distributed continuous variables.

tended to be associated with a better response. The association of HER2 expression with treatment response was not consistent between univariate and multivariable models. Therefore, taken together, our data suggest that the TIIC signature has a greater power for patient stratification (Supplementary Fig. 7b).

### Prognostic use of the proposed TIIC signature in anti-PD-1/PD-L1 immunotherapy

Next, we investigated the prognostic use of the TIIC signature; the univariate Cox proportional hazard regression model was used to calculate the HR of each indicator. Then, we used the HR of each indicator as the weight to multiply the value of the indicator itself and then calculated the weighted sum of the four indicators. In this analysis, we categorised patients into high- and low-score groups based on the TIIC signature. The difference in the survival probability over time between the groups was calculated using the Kaplan–Meier method. As expected, we observed a significant difference in both immune-related progression-free survival (irPFS) and immune-related overall survival (irOS) in the validation cohort (Fig. 5c, d, Supplementary Table 25). Therefore, the TIIC signature might be useful to identify patients that will show active anti-tumour immune responses a priori.

### Discussion

Here, we present a detailed multistep platform for the multi-spectral imaging of tissues that generates high-quality datasets at single-cell resolution and may enable the guidance of precision immunotherapy in GC. Importantly, this approach allowed us to map rare cell types with complex phenotypes, characterise the PD-1 and PD-L1 expression intensity in situ, and assess the value of these parameters and their spatial arrangements as biomarkers.

The current clinical practice recommendations for GC are mainly adapted to the disease stage and the HER2, PD-L1 CPS, EBV and MSI status[5,22]. However, there is a need for more accurate prognostic parameters to guide personalised treatments. For instance, the combination of pembrolizumab, trastuzumab and chemotherapy showed promising efficacy in HER2-positive advanced GC, with an ORR above 74.4% in stage I–III clinical trials, regardless of the PD-L1 status[23,24]. However, the mechanistic basis for the synergy between anti-HER2 and anti-PD-1/PD-L1-based therapies has not been definitively established. Our results showed that the effective scores of exhausted T cells and M1 and M2 macrophages in HER2-negative patients were higher than those in HER2-positive patients, suggesting an unfavourable tumour microenvironment for immunotherapy.

While CD8+ T cell subsets are known to be associated with the mechanism of action of these immunotherapeutic agents, a comprehensive and diverse panel of markers providing comparable prognostic accuracy is desirable for clinical applications. A conflicting prognostic use of CD8+ T cells has been reported, possibly because patients with higher CD8+ T cell density also show higher PD-L1 expression[25,26]. In our study, CD8+ T cells were classified into subcategories. The combination of multiple markers enabled us to identify specific tumour immune cell subsets. We found that a higher density of CD8+PD-1+LAG-3+TIM-3+ T cells and CD68+STING+ macrophages was associated with inferior OS, independently of potential confounding factors. Importantly, these results are consistent with those reported in diffuse large B-cell lymphoma; a high proportion of TIM-3+, LAG-3+ and PD-1+ TIICs translated into poor survival[27]. Differential PD-1 expression in CD8+ T cells is indicative of T cell exhaustion. In fact, the functional analysis of CD8+ T cells in hepatocellular carcinoma showed that the PD-1-high sub-population produced the lowest amounts of tumour necrosis factor (TNF) and/or interferon-gamma (IFN-γ) upon T cell receptor stimulation[28].

In addition, we found that patients with a higher effective density (0–20 μm radius) of CD68+STING+ macrophages, CD68+HLA-DR+CD163− STING+ macrophages and neutrophils showed a significantly shorter OS.

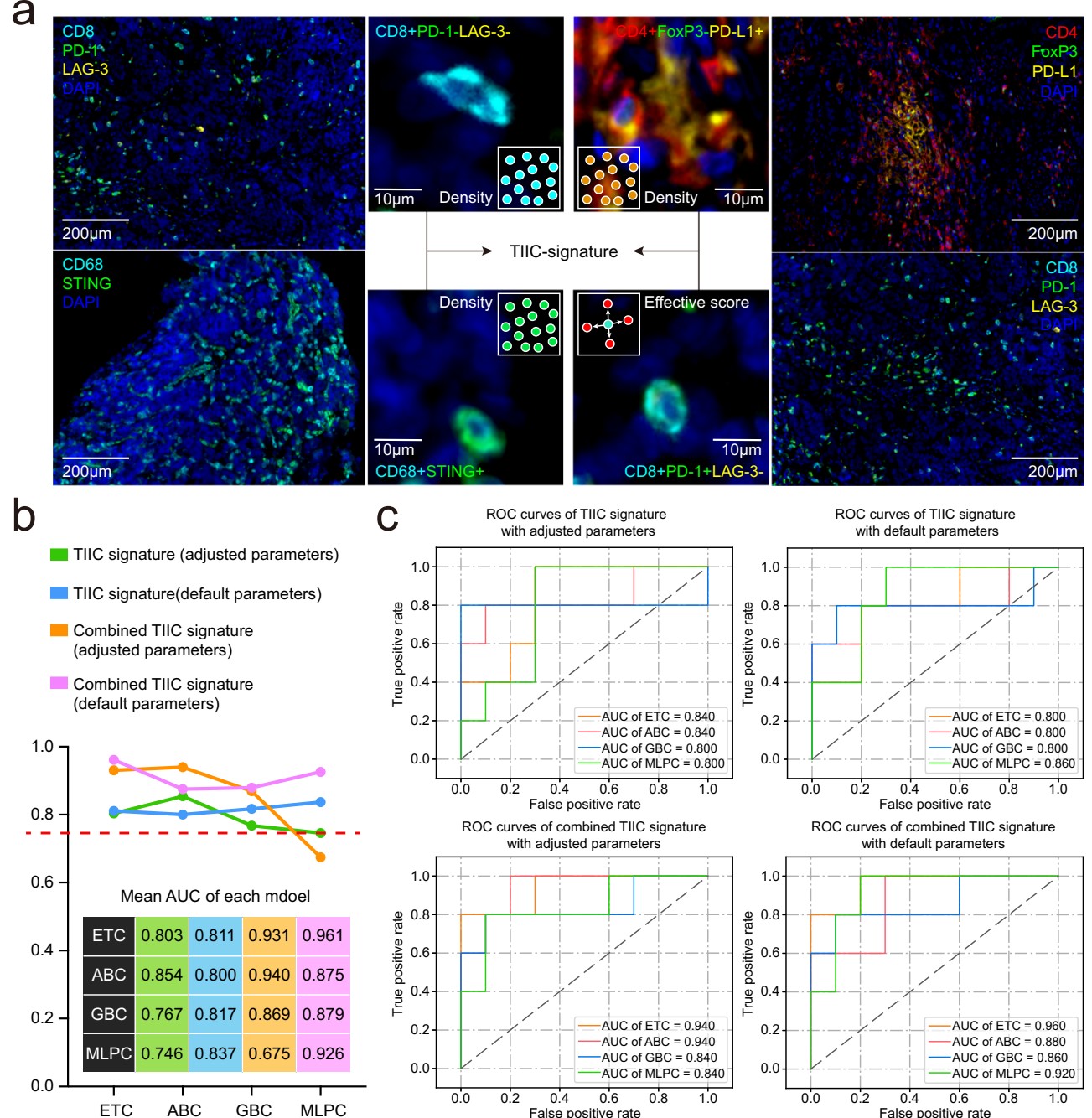

**Fig. 4 | The TIIC signature predicts the response to anti-PD-1/PD-L1-based immunotherapy. a** Definition of the tumour-infiltrating immune cell (TIIC) signature. Red arrows highlight specific immune cells. **b** Average area under the curve (AUC) of TIIC signature and combined TIIC signature (TIIC+ Epstein–Barr virus status + mismatch repair status + PD-L1 combined positive score) in the four machine learning models in the validation cohort. **c** Representative receiver operating characteristic (ROC) curves for the performance of the identified TIIC signature and combined TIIC signature in gastric cancer patients subjected to immunotherapy in the validation cohort. ETC extra tree classifier, GBC gradient boosting classifier, ABC AdaBoost classifier, MLP multi-layer perceptron.

In fact, both the density and effective density of CD68⁺STING⁺ macrophages were associated with inferior survival. These results aligned with those reported in previous studies suggesting that STING or macrophages are negative prognosticators of GC[17,19]. Therefore, our results further validate the negative prognostic value of STING, particularly in the context of defined macrophage subtypes. Altogether, our findings support the concept that a combination of spatial markers enhances the prognostic value in the context of GC.

Our primary aim was to evaluate the density and spatial patterns of TIICs in the context of anti-PD-1/PD-L1 treatment for predicting response to immunotherapy in GC. The density of CD4⁺FoxP3⁻PD-L1⁺ T cells and the effective score of CD8⁺PD-1⁺LAG-3⁻ T cells were associated with a positive response to anti-PD-1/PD-L1 therapy. A higher effective score indicated a higher number of paired immune cells and tumour cells divided by the total number of immune cells across the whole slides. These results are consistent with those of a previous study, showing that a high percentage of CD8⁺PD-1⁺TIM-3⁻LAG-3⁻ cells is correlated with longer median irPFS and higher ORR[29]. Previous reports suggested that *CXCL13* (encoding an effector chemokine) is among the most up-regulated genes in PD-1⁺ tumour-infiltrating cells;

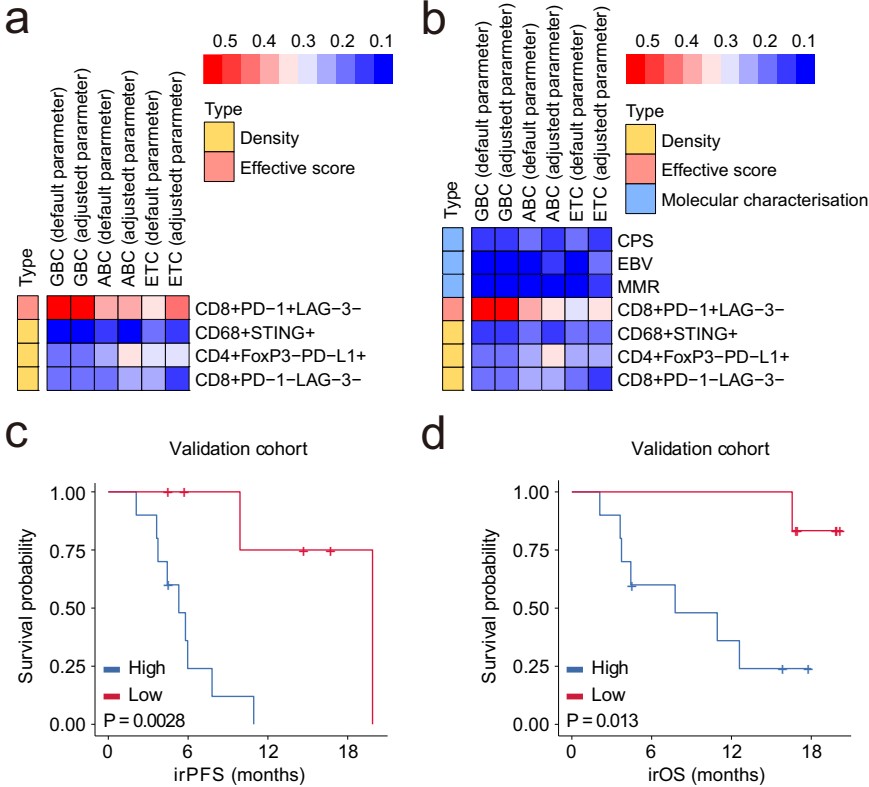

**Fig. 5 | Feature importance of the TIIC signature and predictive value of the TIIC signature in immune-related survival. a, b** The feature importance of each marker in the prediction model. **c, d** Kaplan–Meier curves of the (**c**) immune-related progression-free survival (irPFS) and (**d**) immune-related overall survival (irOS) of anti-PD-1/PD-L1-treated patients stratified by the tumour-infiltrating immune cell (TIIC) signature in the validation cohort. Log-rank (Mantel–Cox) test was used for analysis.

such an up-regulation might underlie the attraction of other immune cell subsets into the tumour tissues[27]. In addition, PD-1+ TILs were found at the tumour–host interface, suggesting that these cells may have an active function in the recruitment of immune subsets into the tumour[30]. Deletion of *CD274* (which encodes PD-L1) in T cells enhances adaptive tumour immunity and activates TAMs, indicating that the use of anti-PD-1/PD-L1 inhibitors will enhance immunity in tumours with a high density of CD4+FoxP3−PD-L1+ T cells[31].

Interestingly, CD8+PD-1−LAG3− T cells and CD68+STING+ macrophages were associated with a negative response to anti-PD-1/PD-L1 therapy. In fact, the CD8+PD-1−LAG-3− T cell type was able to specifically identify a sub-set of patients with a poor immunotherapy response, potentially allowing for the selection of an alternate therapeutic regimen. The high expression of STING in TAMs was associated with poor survival, not only in our study but also in a previous investigation on GC[17]. A functional study showed that knockdown of STING promotes the polarisation of macrophages into the pro-inflammatory subtype, leading to the apoptosis of GC cells. The use of STING antagonists in synergism with PD-1 blockade induced durable anti-tumour immunity with the suppression of peritoneal dissemination of colon cancer and, ultimately, cancer eradication[32]. Since peritoneal metastases are a fatal presentation of gastrointestinal cancers, the same combination therapeutic strategy should be considered to prevent peritoneal carcinomatosis in GC. Collectively, these observations indicated multiple layers of organisation within tumours and effectively predicted the immunotherapy response. Indeed, this study offers further insights into the nature of GC immunity. The information generated can be harnessed for the development and optimisation of effective immunotherapy strategies. Our data clearly show that TIICs have a dominant function in response to ICIs. Importantly, we propose a TIIC signature to predict immunotherapy response.

Our study is not without limitations. Although the m-IHC method used is undoubtedly advantageous over traditional IHC, a consensus set of protein markers for myeloid-derived suppressor cells has not yet been defined, and therefore differences in marker selection will exist. Further studies, including spatial transcriptomics and in vivo and in vitro validation, are essential, with the potential to provide additional biological insights into TIICs in GC. The reason why tumours develop a given TIIC signature is a fundamental question and is worthy of further study. Furthermore, we acknowledge that some immune cell subsets defined by us can be related to one another as 'daughter cells' with distinct phenotypes but derived from the same 'ancestor cell'. Therefore, independent studies are warranted to confirm our findings.

On the contrary, the strengths of the study include the acquisition of the entire tumour microenvironment in all slides, followed by the standardised selection of discrete ROIs. Our study clearly highlights the benefits of computer-assisted cell-density quantitation over the visual assessment of the proportion of positive cells for a given marker by a pathologist alone. Nevertheless, although our results support the prognostic and predictive significance of TIICs in GC, further studies in large cohorts, ideally from prospective clinical trials, are required to confirm our findings. Comparative studies are also needed to show the clear advantage of using the method that we propose here for the evaluation of immune cell infiltrates.

Overall, our results highlight the multi-dimensional marker evaluation of tumour immune infiltrates as a robust and quantitative predictive tool for GC. The exploration of the microenvironment composition of GC samples would offer critical insights into the complex and heterogeneous immune landscape associated tumour progression and immunotherapy treatment response.

## Methods

### Patients and specimens

FFPE GC tissues were obtained from the Department of Pathology, Peking University Cancer Hospital. GC tissues included 80 pre-treatment samples from 80 subjects with histologically confirmed gastric adenocarcinoma diagnosed between July 2014 and December 2019. Among the 60 patients subjected to immunotherapy, 21 were treated with standard-of-care anti-PD-1/PD-L1 antibodies and 39 were treated as part of clinical trials (NCT03472365, NCT03713905). We excluded patients with concurrent autoimmune diseases, HIV, or syphilis. This study was approved by the Ethics Committee of the Peking University Cancer Hospital (NCT03472365, NCT03713905)[33,34]. All participants or their legal guardians signed informed consent forms. EBV status was determined using in situ hybridisation with probes against Epstein–Barr encoded RNA 1 (EBER1). Additionally, the MMR status was assessed using IHC analysis of the expression of the DNA mismatch repair proteins MLH1, MSH2, MSH6 and PMS2, as previously described[35]. CPS was defined as the number of PD-L1-positive tumour cells (partial or complete membrane staining), lymphocytes and macrophages (membrane staining or intracellular staining, or both) divided by the total number of viable tumour cells multiplied by 100[2]. Responders were defined as patients with a RECIST complete response (CR) or partial response (PR), while non-responders were defined as those with progressive disease (PD) or stable disease (SD). OS was defined as the period from diagnosis to the time of death or end of follow-up, whichever occurred first. irOS was defined as the time from initial immunotherapy to the day of death or the end of follow-up, whichever occurred first. irPFS was defined as the time from initial immunotherapy to the day of disease progression, death or the end of follow-up, whichever occurred first.

### Multiplex immunohistochemistry

Multiplex IHC staining was performed to visualise the expression of CD8, PD-1, TIM-3, LAG-3, CD4, FoxP3, CTLA-4, PD-L1, CD68, CD163, HLA-DR, STING, CD20 and CD66b in tumour tissues in four panels. The specimens were collected within 30 min after tumour collection and fixed in formalin for 24–48 h. Dehydration and paraffin embedding were performed using routine methods. Five consecutive sections (4 μm-thick) were cut from paraffin blocks. One section was used for H&E staining. Four FFPE tumour slides (4 μm) were melted at 60 °C for dehydration for 12 h. Paraffin sections were de-paraffinised in xylene and re-hydrated in alcohol. Heat-induced antigen retrieval was performed in ethylenediaminetetraacetic acid (EDTA) buffer, pH 9.0 (or citrate buffer, pH 6.0, for FoxP3 staining) using a microwave oven. The sections were blocked with commercially available blocking buffer (Dako, Santa Clara, CA; cat. X0909) for 10 min. The primary antibodies used for each staining are listed in Supplementary Table 26. The concentration and staining order of the antibodies used in this study were optimised in advance. The slides were serially incubated with primary antibodies and horseradish peroxidase-conjugated secondary antibodies (Biolynx, Hangzhou, China, cat. BX10001) and subjected to tyramide signal amplification (TSA). After each round of TSA operation, the slides were heated for antigen retrieval and antibody stripping. After all sequential staining steps, the cell nuclei were stained with 4′,6-diamidino-2-phenylindole (DAPI, Sigma-Aldrich, St. Louis, MO; cat. D9542)[36]. Two specialist pathologists (blinded to the patient's information) evaluated all GC specimens.

### Multi-spectral imaging

Images were acquired using the Mantra Quantitative Pathology Imaging System (PerkinElmer, Waltham, MA). The multi-spectral images were visualised in a Phenochart. Briefly, representative ROIs were chosen by two specialist pathologists, and multiple fields of view were acquired at 20× for further analysis. The ROIs were defined as follows: normal tissue adjacent to the tumour (N), the area within the specimen but not within the tumour; IM, the area at the interface of tumour and normal tissues (approximately 1–1.5 mm; depth defined by the size of the microscopy field); TC, -the tumour centre ROIs were selected with fixed-size stamps in a Phenochart (PerkinElmer), based on the previously acquired whole-slide scan images. A 1 × 1 (930 × 700 μm; ×20 object lens) stamp was used. As many viable regions as possible in each specimen were selected with minimal overlap. All processed data were subjected to quality control (QC) by a pathologist, with the subsequent exclusion of the inappropriate regions from the analysis as well as the confirmation of outlier results.

### Imaging data analysis

The multi-spectral images were analysed using the inForm image analysis software 2.4 (PerkinElmer, Waltham, MA). The spectral library was built based on the single-stained slides for each fluorophore, and unstained sections were used to extract the auto-fluorescence spectrum of the tissues. The inForm software actively learned the phenotyping algorithm from all spectrally unmixed images. Each DAPI-stained cell was individually identified according to its combination of fluorophore characteristics and cell morphology features associated with a segmented nucleus (DAPI signal).

The acquired images ($n = 6488$) were analysed with inForm for tissue-component segmentation of tumour cell and stroma regions and cell phenotyping. The density of cells in each ROI was calculated via the normalisation of the total cell counts by the total area (cell/mm$^2$). The TC fields from a single patient were used as a cluster. In each patient, we calculated the density of cells in TC, IM and N by the cluster, including all ROIs from the TC, IM, N of this patient, respectively.

### Construction of the machine learning models

Supervised machine learning methods were used to train classifiers to map the characteristics of a given training cohort to a learning target, which in this research was the immunotherapy response. The performance of the learned model was then evaluated on the validation cohort. The supervised classifiers in this research were constructed using scikit-learn (version 0.23.2), one of the most popular machine learning programs in Python[37]. Four types of ensemble classifiers, including ETC, GBC, ABC and MLP, were built. For each type of classifier, we selected the appropriate hyper-parameters through a grid-search and adopted a 3-fold cross-validation to improve the robustness of the model. The detailed information of the candidate parameters is presented in Supplementary Table 27. The classifiers originating from the training cohort were applied to the validation cohort, and the AUC of each classifier was calculated. To compare the effect of the adjustment of the hyper-parameters on model efficacy, we also tested models that used default parameters.

The sample was randomly divided into three groups in 3-fold cross-validation. Some classifiers (such as ETC) had random starting points or branches in the execution process. Therefore, we performed 5000 repetitions of the whole prediction process and presented the average AUC of 5000 predictions to represent the model efficacy, which might reduce the impact of sampling error on the evaluation of the model.

One patient in the validation cohort had an absent ORR; we excluded this patient in the evaluation of treatment response but included this patient in survival analysis.

### Statistics and reproducibility

The relationships between TIICs and clinico-pathologic features were evaluated using the Mann–Whitney $U$ test, the Kruskal–Wallis test, or Pearson correlation analyses, as appropriate. For multiple comparisons of immune cell density and tumour location (TC, IM, N), we used Dunn's adjustment[38]. We also used the Kaplan–Meier method to estimate survival functions and the log-rank test to compare survival distributions. We conducted logistic regression analyses to examine

the association of TIICs or molecular features of GC with treatment response. The assumption of proportionality of hazards was assessed using a time-varying covariate in the Cox models with a cross-product term of survival time and each TIIC. The proportional hazard assumptions were generally satisfied for survival ($P > 0.05$). In addition, to disclose the potential relationship between each TIIC and GC survival, multi-variable-adjusted Cox proportional hazards regression analysis was used[39]. All statistical analyses were performed using SPSS 26.0 (IBM, Armonk, NY, USA), GraphPad Prism 7.0 (GraphPad Software, San Diego, CA, USA), and software package R (version 4.1.2). All *P*-values were two-sided.

To test reproducibility, we assessed the densities of four cell types (CD4+T cells, CD8+T cells, B cells and macrophages; represented by CD4, CD8, CD20 and CD68, respectively) in six samples (six sections for each case) to test intra-patient variance. The overall densities of these cells and the coefficient of variation among six sections were generally similar (Supplementary Fig. 8). The coefficient of variation suggests that the m-IHC staining and ROI selection were highly consistent in our cohort.

### Reporting summary
Further information on research design is available in the Nature Research Reporting Summary linked to this article.

## Data availability
All other data are available in the article and its Supplementary files or from the corresponding author upon reasonable request. Source data are provided with this paper.

## Code availability
All R packages used in this study are available from CRAN (https://cran.r-project.org/web/packages/available_packages_by_name.html) or Bioconductor (https://www.bioconductor.org/). Machine learning models were constructed using the scikit-learn package, which is available online (https://scikit-learn.org/stable/).

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

## Acknowledgements

This work was supported by National Natural Science Foundation of China (91959205 to L.S., 12090022 to B.D., 11831002 to L.Z.), Beijing Natural Science Foundation (7222021 to Y.C., Z180001 to B.D.), the Capital Foundation of Medical Development (2020-1-1022 to XT.Z.), Wu Jieping Medical Foundation (320.6750.2021-02-15 to Y.C.), and Innovation Fund for Outstanding Doctoral Candidates of Peking University Health Science Center (BMU2021BSS001 to KR.J.). The funder had no role in study design, data collection and analysis, decision to publish, or preparation of the manuscript. The authors thank Yingcheng Wu (Zhongshan Hospital, Fudan University) as well as Kang Liu and Jing Huang (Tsinghua University) for providing valuable suggestions. The authors are also grateful to Lei Jiao, Jie Ma, Xiaojiao Hao and Pengxiang Li (Panovue Biotechnology [Beijing] Co., Ltd) for providing technical assistance.

## Author contributions

L.S. and Y.C. conceived and supervised the study. Y.C., K.J., Y.L. and YY.L. contributed to sample collection and collection of patient clinical information. Y.S. and Y.H. contributed to pathology review. J.Y. reviewed the CT images. K.J., L.Z., ZF.C., J.Z. and B.D. supervised the bioinformatics data analysis, data integration and interpretation. Y.C. and K.J. contributed to data processing, integrative analyses, and generating figures and tables. XW.Z., C.Z., XT.Z., YL.L., J.G. and J.L. assisted with data processing and analysis. Y.C. wrote the manuscript. All authors revised the manuscript.

## Competing interests

The authors declare no competing interests.
