## [Peer Review File · Nature Communications]

Predicting response to immunotherapy in gastric cancer via multi-dimensional analyses of the tumour immune microenvironmentREVIEWER COMMENTS

Reviewer #1 (Remarks to the Author):

In their original work the authors attempt to address an important clinical question in gastric cancer; the development of better predictive and prognostic biomarkers in immune checkpoint inhibition. Clinically the data is quite clear that increasing PD-L1 expression, assessed by the CPS method, is associated with increased chance of response and greater magnitude of benefit with PD-1 monotherapy and combination with chemotherapy. To try to address this the authors conduct a multiplex IHC analyses of tumor-infiltrating immune cells (TIIC) and integrate the quantitative immune cell data with spatial organization (proximity to tumor cells) in cohorts of clinically annotated samples. Using a machine learning models they propose TIIC signatures capable associated with differential outcomes. The strengths of the work include the well collected and annotated cohorts, the relatively robust IHC determinations, and the incorporation of some more automated pathology analyses. Limitations include that other similar types of multiplex analyses are published, including in GC, small sample sizes among some of the cohorts (lower PD-L1 expression, EBV, etc), and limited biologic insights. There is also substantial literature on other signatures predicting response, including the validated IFN-gene signature but clinical implementation has not been feasible despite the improved predictive ability. I would ask the authors to address the following comments;

Major:

- 1. The authors repeatedly state the algorithm (TIIC signature) predicts response to PD-1 based immunotherapy but most assessments are based on overall survival. Have they done work to show that the TIIC signature is predictive of response rate to PD-1? Does this data exist in the cohorts? Progression-free survival is perhaps a better measure of the effect of a given therapy than overall survival which is affected by subsequent therapies. Could the TIIC signature be more prognostic than predictive? This should be addressed more clearly. Also the data seems to be presented as PFS and OS but was this irPFS and irOS?**
- 2. Table S19 does not clearly describe the association between immune cells with response rate and is this intended to be responder versus non-responder? If rate is intended then reporting the ORR in overall cohort would be helpful and also providing the number of patients In each group.**
- 3. It actually appears that PD-L1 CPS scores perform relatively well on their own in the cohort. In looking at the inputs into the multivariable Cox model is does not include the PD-L1 CPS score or the molecular tumor information (MSI, EBV, etc) both of which are known independent prognostic/predictive variables. Why were these variable not included? This leads to a question of how much does the TIIC signature add beyond PD-L1 CPS scoring and would the TIIC signature AUC improve if PD-L1 score and/or molecular tumor information was added?**
- 4. The inclusion of a small number of cases with serial samples (n = 5) pre and post treatment is interesting but unclear what this adds to development of the TIIC signature or annotation to outcomes. I would consider removing this or more clearly stating how it helps in developing the TIIC signature**
- 5. Line 266-268 does not appear supported by the data. The authors do show a change in several TIIC populations but how this delta can be used to predict response is not clear. Suggest tempering this statement**
- 6. The discussion could be condensed and reads somewhat repetitive with the results section in the current version. In addition to the need to validation in larger cohorts, ideally from prospectively collected clinical trials, would the authors plan for other methods like spatial transcriptomics that may allow for more biologic insights? Why do some tumors develop a given TIIC signature over another? These are important and more fundamental questions that may be worthy of mention.**

Minor:

1. The discussion could be condensed and reads somewhat repetitive with the results section in the current version.
2. Supplemental Table S21: "Referent" should be changed to "Reference" throughout
3. Supplementary figure 5d is interesting and would consider movement to main text/figures.

Reviewer #2 (Remarks to the Author):

In this manuscript, the authors report the use of multi-dimensional tumor-infiltrating immune cells (TIICs)-signature to predict survival in patients with gastric cancer (GC) treated with immunotherapy. They show that the density of the specific immune population and the effective score considering spatial organization are associated with response to anti-PD-1/PD-L1 therapy and could be used as a predictive biomarker for GC patients. This is an exciting study highlighting the multi-dimensional marker evaluation of TIICs-signature as useful tool for the prediction of the prognosis and selection of patients most likely to benefit from immunotherapy. The results are clearly described, the manuscript is well-written, and the approach may provide a new therapeutic strategy for GC. The evaluation of TIICs using multi-dimensional analyses in this context is an innovative approach. However, there are multiple limitations and clarifications that need to be addressed.

Specific comments:

1. The authors described the cellular phenotypes in the GC tumor microenvironment and new predictive value for the survival prognosis and response to immunotherapy using multi-dimensional TIICs-signature. However, mechanistic insights are lacking. The authors do not provide any molecular mechanism assessment with in vivo/vitro validation.
2. In general, the authors focused on the association between the density, the spatial organization of several immune subsets and association with clinical outcomes. Considering the functional interaction between them, the authors need to assess not only exhaustion and immune checkpoint marker but also the spatial organization of other immune subsets such as TRM, TEF, and activation status of CD8+ T cells and NK cells, which might also be important.
3. It looks like the approach was to use one section from each tumor and image multiple fields within that section. Although this is not clear from the write-up, the sampling is probably from biopsy specimens. Since it is well-known that the immune cells and markers are highly heterogeneous within individual tumors, it is important to look at multiple sections from each tumor (at the least). Currently there is a high risk of sampling bias. Maybe an intra-patient variance (measured from multiple sections of each specimen) for the immune cell subtypes is needed.
4. Assessment of high/low density of subtypes (e.g., DC68+STING+) will be skewed towards the tumors with higher number of imaged fields. Approximately 6,500 fields were used from 80 specimens in the machine learning model. Was each field used as a separate sample or were all fields from a single tumor used as a cluster? How was this defined in the model?
5. Are the training and validation cohort sizes large enough (considering 13 immune markers were used) for a meaningful comparison of outcomes? Please provide references if they are any.
6. Not sure how commonly the protocol for IHC staining of 13 markers in one tissue section is used in practice. Are there any artefacts from the previous staining on the new ones? I'm assuming the last few staining will have significant artefacts considering some antibodies were used at a very high concentration.
7. The authors adopted 20µm radius to calculate the effective density as an additional measurement. And, it is said that it was pre-selected and consistent with the previous reports. Please provide more detailed information and references in the methods.
8. What is the dominant predictive biomarker value of the density versus spatial

organization for OS and response to immunotherapy?

9. Page 4, in the results section, the authors state that 80 patients were enrolled in this study between July 2014 and December 2019. On the other hand, in the methods part, it is described that GC tissues included 85 pre-treatment samples from 80 subjects with histologically confirmed gastric adenocarcinoma collected from March 2018 to December 2020.

10. Page 9, line184, the authors describe M1 macrophages as CD68+CD163+HLA-DR-, and M2 macrophages as CD68+HLA-DR+CD163-. However, some other reports describe M2 macrophages as CD163+cells. Please address this discrepancy.

11. It would be useful to describe why the stage I/II/III patients enrolled in this study were not eligible for other treatments such as surgery.

Reviewer #3 (Remarks to the Author):

The study employs a newly developed multiplex immunohistochemistry approach to provide insight in the location of immune populations in gastric cancer to explore their potential to predict response to PDL1/PD1 immunotherapy. Based on their findings, a tumor-infiltrating immune cell signature was compiled out of the density of CD4+FoxP3-PD-L1+ T cells, CD8+PD-1-LAG3- T cells, and CD68+STING+ macrophages, and the effective score of CD8+PD-1+LAG3- T cells. Analysis revealed that this signature had predictive value for response to PDL1/PD1 therapy and a priori anti-cancer immune responses. Finally, the immune composition was studied before and after PD1/PDL1 therapy to show that CD8+PD1- cells increased, but STING+ macrophages and PD1/PDL1+ T cells decreased after therapy.

This manuscript currently remains very inaccessible in its presentation. 1. The immunohistochemistry data is hard to read, not just because of poor resolution, but also because no effort has been made to point out the relevant cells in the pictures of Fig 1c-f and 5c. It would perhaps also be helpful to use color coding to re-label double or triple positive cells for clarity. 2. The heatmap presentation in Fig 2c and 3c is only directly readable for association with the combined positive score (CPS) and not for any of the other tumor indicators. CPS is also not clearly defined. 3. The presence of a large number of associations of immune cells to other tumor parameters throughout the text actually diffuse the key message of the paper, given that many of them do not directly relate to the prognostic value for PD1/PDL1 therapy. These statements should be placed in this context or otherwise removed. 4. The proximity of immune cells to the tumor cells was studied to investigate whether immune cells in close range to tumor cells would be more beneficial or more detrimental for the patient. Impact is observed for macrophages at close range on patient survival and for PD1+ CD8 T cells for treatment response to PD1/PDL1 therapy. This latter finding appears important for the key message of the paper but yet only the data on macrophages has been displayed in main figure 3. 5. The definition of the tumor-infiltrating immune cell signature actually obscures how the parameters associate with predictive value for PD1/PDL1 therapy. This omission should be clarified.

Reviewer #4 (Remarks to the Author):

MAJOR COMMENTS

1. Line 229: The authors applied four different algorithms to predict response of gastric cancer to immunotherapy and report the maximum and minimum area under the receiver operating characteristic curve. However, there are only 15 subjects in the validation set because among the 16 in the validation set, one was missing ORR. Therefore, the 95% CIs should be reported for each algorithm rather than the point estimates of the AUC for the 4 algorithms. Also, lines 230-232 describe which methods perform better. It would be better if for each of the four methods the AUC and 95% CI were reported rather than simply a min-max range.

2. Lines 213-215: The authors wanted to ensure balance between the training and validation sets. One patient was excluded from the validation set (see lines 476-477). It would seem more appropriate if Table 2 compared the 15 patients actually used in validation rather than including one patient that was subsequently omitted.
3. Lines 236-239: The authors should report the AUC for tumor mutation burden, PD-L1 expression, microsatellite instability, and Epstein-Barr virus infection status and a multivariable model for these given they have been proposed to identify susceptibility to PD-1/PD-L1 inhibitors. What about HER2 (omitted from this section)? That is, the authors should demonstrate their biomarkers offer some advantage over current ones. It does not seem to be a fair comparison to combine their biomarkers but analyze the others univariately.
4. Throughout: The expression of 16 proteins was assessed along with the density and spatial location of immune cells. Tumor core (TC) [N=4477], invasion margin (IM) [N=993], and peritumoral normal areas (N) [N=1018] were reported as regions of interest. Despite this, the authors reported p-values with seemingly no adjustment for multiple hypothesis tests. The authors need to use some adjustment for multiple comparisons or alternatively report the false discovery rates instead of the raw p-values. Also, throughout the manuscript and figures, "ns" should be replaced with the appropriate p-value or FDR.
5. Lines 242-257: When deriving the high- vs low-score groups, it is unclear whether Cox PH models were fit to a training dataset or to the entire cohort of patients. Based on Figures 2/3/4, it seems the latter was used. If indeed the entire cohort was used, the scores need to be re-estimated by fitting models only to the training data and then applying those score to the validation data. Only the validation results are of intrinsic interest. Additionally, there is no utility in presenting results for the combined training+validation sets.

MINOR COMMENTS

6. Figure 2: It would be helpful if Tumor core (TC), invasion margin (IM), and peritumoral normal areas (N) were added to the caption.
7. Figure 4: The ROC for the training data is not of intrinsic interest.
8. The title, "Predicting response to immunotherapy in gastric cancer...", may not be entirely reflective of the manuscript's content given 20 of the 80 gastric cancer patients did not receive immunotherapy. Survival was assessed in all 80.
9. According to lines 73-74, five of the 80 patients contributed both pre- and post-treatment samples. However, lines 389-390 state, "GC tissues included 85 pre-treatment samples from 80 subjects." These statements seem to be conflicting.
10. Lines 118-121: The authors tested for whether there was a significant difference between groups but then state, "In general, patients showed similar densities of TIICs according to the Lauren classification, tumor differentiation, and tumor location." Absence of evidence is not evidence of absence. This needs to be restated that generally, there are few significant differences between Lauren classification, tumor differentiation, and tumor location with respect to densities of TIICs.
11. Lines 124-125: Why did the authors exclude CD4+, CD4+FoxP3-, and CD68+HLA-DR+CD163- from this sentence?
12. Lines 127-128: Why did the authors exclude CD68+CD163+HLA-DR-STING+ from this statement?
13. Lines 128-130: Why did the authors exclude CD4+, CD4+FoxP3-, and CD68+STING+?
14. Lines 130-131: It is unclear how "combined positive score" is derived to then come up with the four groups. Depending on how it was formed, it may or may not make sense to report Supplementary Table S8.
15. Lines 143-145: The interpretation that the prognostic value was significantly adjusted is not correct. The authors need to reword this sentence to reflect that those tumor-infiltrating T cell subsets were significantly associated with overall survival.
16. Lines 181-187: It is not clear how effective scores were assigned and whether they are continuous and meet parametric assumptions in order for Student's t-test and ANOVA are appropriate.
17. Line 404: The authors should clearly articulate what subjects were censored when analyzing immune-related progression-free survival.

- 18. Lines 464-465: The authors should cite relevant references for each method.**
- 19. Lines 466-467: For each method, the authors should state what hyperparameters were tuned.**
- 20. Lines 473-475: First, it is unclear what parts of the process were repeated 5000 times. If repeated 3 fold-CV did not also include variable selection, the results will not reflect generalization performance.**
- 21. Lines 485-486: To examine the potential relationship between "each" TIIC and GC survival, the authors should fit univariable not multivariable Cox PH models. Also, when fitting Cox PH models the authors should verify the PH assumption and also when fitting multivariable Cox PH models, the authors need to disclose their process for arriving at a final model and then checking model fit.**
- 22. Throughout: Many paragraphs start with something like "Based on the above results." When in print, those results may not be "above" so these phrases should be reworded.**

REVIEWER COMMENTS

Reviewer #1 (Remarks to the Author):

In their original work the authors attempt to address an important clinical question in gastric cancer; the development of better predictive and prognostic biomarkers in immune checkpoint inhibition. Clinically the data is quite clear that increasing PD-L1 expression, assessed by the CPS method, is associated with increased chance of response and greater magnitude of benefit with PD-1 monotherapy and combination with chemotherapy. To try to address this the authors conduct a multiplex IHC analyses of tumor-infiltrating immune cells (TIIC) and integrate the quantitative immune cell data with spatial organization (proximity to tumor cells) in cohorts of clinically annotated samples. Using a machine learning models they propose TIIC signatures capable associated with differential outcomes. The strengths of the work include the well collected and annotated cohorts, the relatively robust IHC determinations, and the incorporation of some more automated pathology analyses. Limitations include that other similar types of multiplex analyses are published, including in GC, small sample sizes among some of the cohorts (lower PD-L1 expression, EBV, etc), and limited biologic insights. There is also substantial literature on other signatures predicting response, including the validated IFN-gene signature but clinical implementation has not been feasible despite the improved predictive ability. I would ask the authors to address the following comments;

Major:

Comment: “1. The authors repeatedly state the algorithm (TIIC signature) predicts response to PD-1 based immunotherapy but most assessments are based on overall survival. Have they done work to show that the TIIC signature is predictive of response rate to PD-1? Does this data exist in the cohorts? Progression-free survival is perhaps a better measure of the effect of a given therapy than overall survival which is affected by subsequent therapies. Could the TIIC signature be more prognostic than predictive? This should be addressed more clearly. Also the data seems to be presented as PFS and OS but was this irPFS and irOS?”

Reply: We appreciate your valuable comment. We have shown that the TIIC signature is predictive of response rate to PD-1/PD-L1. Please see the results of TIIC signatures with patients' response rates to PD-1/PD-L1 in Figure 4b-c and Supplementary Figure 7a. We presented the ROC curves and mean AUC of four machine learning models of the predictive value of the TIIC signature in patients treated with anti-PD-1/PD-L1 therapy. Furthermore, we presented the association of the TIIC signature with irPFS and irOS in gastric cancer patients who received immunotherapy in the validation cohort in Figure 5c-d. As we can see, the TIIC signature harbors both prognostic and predictive value. We are sorry about the confusion. The OS of 80 patients is presented in Figure 2 and 3, and irPFS and irOS is presented in Figure 5.

Comment: “2. Table S19 does not clearly describe the association between immune cells with response rate and is this intended to be responder versus non-responder? If rate is intended then reporting the ORR in overall cohort would be helpful and also providing the number of patients in each group.”

Reply: Thank you very much for your suggestion. In the methods section, we defined responders as patients with a RECIST complete response (CR) or partial response (PR), while non-responders were defined as those with progressive disease (PD) or stable disease (SD). Table S19 intended to describe the association between immune cells with response [responder (PR+CR) vs. non-responder (SD+PD)]. Thus, we reported the ORR in the training cohort. As you requested, we have provided the number of patients in each group. Please kindly find the revision in Supplementary Table S19.

Comment: “3. It actually appears that PD-L1 CPS scores perform relatively well on their own in the cohort. In looking at the inputs into the multivariable Cox model is does not include the PD-L1 CPS score or the molecular tumor information (MSI, EBV, etc) both of which are known independent prognostic/predictive variables. Why were these variable not included? This leads to a question of how much does the TIIC signature add beyond PD-L1 CPS scoring and would the TIIC signature AUC improve if PD-L1 score and/or molecular tumor information was added?”

Reply: Thank you for your helpful suggestion. First, we included the PD-L1 CPS score, MSI status, and EBV status into the multivariable Cox model as you requested. The prognostic value of the TIIC signature was still significant after adjustment using the multivariate Cox model. Please see the revision in Supplementary Table S24.

Second, we added PD-L1 CPS, MMR status, and EBV status in the machine learning models. The combined TIIC signature (TIIC+CPS+MMR+EBV) had a better AUC in the ETC, GBC, and ABC models, but not in the MLP classifier. Please see the revision in Results section, Figure 4b-c, and Supplementary Tables S20-21.

Third, we also quantified the contribution of each marker in the prediction through feature importance using the scikit-learn package (Supplementary Table S22-23). We outputted the feature importance and the average value of each parameter to present its contribution. As presented in Figure 5b, the effective score of CD8⁺PD-1⁺LAG-3⁻ had a higher feature importance than the density of CD68⁺STING⁺, CD4⁺FoxP3⁻PD-L1⁺, and CD8⁺PD-1⁻LAG-3⁻ cells, EBV and MMR status, and PD-L1 CPS. Thus, the dominant predictive marker is spatial organization for response to immunotherapy.

Comment: “4. The inclusion of a small number of cases with serial samples (n = 5)

pre and post treatment is interesting but unclear what this adds to development of the TIIC signature or annotation to outcomes. I would consider removing this or more clearly stating how it helps in developing the TIIC signature.”

Reply: We appreciate your comment. The contribution of 5 serial samples is limited in the current project. As you suggested, we removed serial samples (n = 5) pre- and post-treatment.

Comment: “5. Line 266-268 does not appear supported by the data. The authors do show a change in several TIIC populations but how this delta can be used to predict response is not clear. Suggest tempering this statement.”

Reply: Thank you for your suggestion. The current data of 5 serial samples could add limited information to the whole picture. As you suggested, we removed Figure 5 and the relevant statement.

Comment: “6. The discussion could be condensed and reads somewhat repetitive with the results section in the current version. In addition to the need to validation in larger cohorts, ideally from prospectively collected clinical trials, would the authors plan for other methods like spatial transcriptomics that may allow for more biologic insights? Why do some tumors develop a given TIIC signature over another? These are important and more fundamental questions that may be worthy of mention.”

Reply: We appreciate your great suggestion. We condensed the discussion section and revised accordingly. Please see the condensed discussion.

As you suggested, we also revised the Discussion section as follows: “Further studies including spatial transcriptomics and *in vivo* and *in vitro* validation are essential, with the potential to provide additional biologic insights into TIICs in GC. The reason why tumors develop a given TIIC signature is a fundamental question and is worthy of further study.”

Minor:

Comment: “1. The discussion could be condensed and reads somewhat repetitive with the results section in the current version.”

Reply: We appreciate your great suggestion. We condensed the discussion section and revised accordingly. Please see the condensed discussion.

Comment: “2. Supplemental Table S21: “Referent” should be changed to “Reference”

throughout.”

Reply: Thank you for correcting the description. We have revised this accordingly. Please see the revision in Supplemental Table S24.

Comment: “3. Supplementary figure 5d is interesting and would consider movement to main text/figures.”

Reply: We appreciate your suggestion. We made substantial changes throughout the manuscript. As we mainly focused on the tumor immune environment, we established a multi-dimensional predicting model for the treatment response of immunotherapy. The TIIC signature we propose mainly includes immune checkpoint markers and exhaustion status of CD8+ T cells to identify patients who could benefit from immunotherapy. Eventually, we established a TIIC score, which could support our conclusion very well. Chemotherapy is not our main hypothesis in this work, and as such, we are sorry we could not move this to the main text. Thank you again for all your valuable suggestions to the manuscript.

We thank you for all the constructive comments. We hope you will find that the manuscript has been significantly revised and the concerns have been thoroughly addressed. Thank you again for your time and effort.

Reviewer #2 (Remarks to the Author):

In this manuscript, the authors report the use of multi-dimensional tumor-infiltrating immune cells (TIICs)-signature to predict survival in patients with gastric cancer (GC) treated with immunotherapy. They show that the density of the specific immune population and the effective score considering spatial organization are associated with response to anti-PD-1/PD-L1 therapy and could be used as a predictive biomarker for GC patients. This is an exciting study highlighting the multi-dimensional marker evaluation of TIICs-signature as useful tool for the prediction of the prognosis and selection of patients most likely to benefit from immunotherapy. The results are clearly described, the manuscript is well-written, and the approach may provide a new therapeutic strategy for GC. The evaluation of TIICs using multi-dimensional analyses in this context is an innovative approach. However, there are multiple limitations and clarifications that need to be addressed.

Specific comments:

Comment: “1. The authors described the cellular phenotypes in the GC tumor microenvironment and new predictive value for the survival prognosis and response to immunotherapy using multi-dimensional TIICs-signature. However, mechanistic insights are lacking. The authors do not provide any molecular mechanism assessment with *in vivo/vitro* validation.”

Reply: We appreciate your valuable comment. In this study, we mainly focused on the tumor-immune environment and established a multi-dimensional predicting model for the treatment response of immunotherapy. This will be essential for multiplex imaging technologies to realize their potential as biomarker discovery platforms and ultimately as diagnostic tests for clinical therapeutic decision-making.

In the Discussion, we cited several previous mechanistic studies from which we could assume that the cellular phenotypes of TIIC signatures were related to immunotherapy. The lack of *in vivo* and *in vitro* validation is our limitation. We added this statement in the Discussion section.

Comment: “2. In general, the authors focused on the association between the density, the spatial organization of several immune subsets and association with clinical outcomes. Considering the functional interaction between them, the authors need to assess not only exhaustion and immune checkpoint marker but also the spatial organization of other immune subsets such as TRM, TEF, and activation status of CD8+ T cells and NK cells, which might also be important.”

Reply: We appreciate your great suggestion. First, we totally agree that TRM, TEF, and activation status of CD8+ T cells and NK cells are important immune subsets. In

this study, our primary aim was to evaluate the density and spatial patterns of TIICs for a better understanding of the determinants of response to immunotherapy in GC. As we have limited FFPE blocks to obtain sections in patients, the markers stained in this study were pre-selected based on previous studies showing that these markers may be associated with the efficacy of anti-PD-1/PD-L1 immunotherapy.^{1,2} Thus, the TIIC signature we propose mainly includes immune checkpoint markers and exhaustion status of CD8+ T cells to identify patients who could benefit from immunotherapy. Eventually, we established a TIIC score which could support our conclusion very well. Unfortunately, we did not have enough sections for additional marker examinations in this cohort. We are planning to evaluate further immune subsets including TRM, TEF, and activation status of CD8+ T cells and NK cells in a future project. Thank you very much for your great suggestion.

Comment: “3. It looks like the approach was to use one section from each tumor and image multiple fields within that section. Although this is not clear from the write-up, the sampling is probably from biopsy specimens. Since it is well-known that the immune cells and markers are highly heterogeneous within individual tumors, it is important to look at multiple sections from each tumor (at the least). Currently there is a high risk of sampling bias. Maybe an intra-patient variance (measured from multiple sections of each specimen) for the immune cell subtypes is needed.”

Reply: We apologize for the confusion. First, we used five consecutive sections (4 μm) for each sample (one section for hematoxylin and eosin, four sections for multiplex immunohistochemistry). We revised this in the methods section.

Second, tumor heterogeneity is a common problem in all varieties of cancers. In advanced gastric cancer, biopsy is widely used in clinical and translation studies including TCGA analysis.^{3,4} In our cohort, 42 samples were from surgery and 38 samples were from endoscopic biopsy. A minimum of 5 tumor-containing biopsies is required to confirm HER2 expression and was the standard used in our cohort, which is suggested in treatment guidelines.⁵

Third, as you requested, we tested intra-patient variance. We assessed the densities of 4 major cell types (CD4+ T cell, CD8+ T cell, B cell, and macrophage; represented by CD4, CD8, CD20, and CD68, respectively) in 6 samples (6 sections for each sample) (as shown in Figures 1-6). The overall densities of these cells were generally similar for inter-patient measurements (as shown in Supplementary Figure S8). We presented the coefficient of variation in Supplementary Figure S8. The coefficient of variation suggested that the mIHC staining and ROI selection maintained a high consistency, which can rule out the influence of heterogeneity on our conclusions.

Patient 1

Section 1

Section 2

Section 3

Section 4

Section 5

Section 6

Figure 1. Representative composite and single-stained images of the multiplex IHC panel in 6 sections in Patient 1. Scale bar: 80 µm.

Patient 2

Section 1

Section 2

Section 3

Section 4

Section 5

Section 6

Figure 2. Representative composite and single-stained images of the multiplex IHC panel in 6 sections in Patient 2. Scale bar: 80 µm.

Patient 3

Section 1

Section 2

Section 3

Section 4

Section 5

Section 6

Figure 3. Representative composite and single-stained images of the multiplex IHC panel in 6 sections in Patient 3. Scale bar: 80 μ m.

Patient 4

Section 1

Section 2

Section 3

Section 4

Section 5

Section 6

Figure 4. Representative composite and single-stained images of the multiplex IHC panel in 6 sections in Patient 4. Scale bar: 80 μ m.

Patient 5

Section 1

Section 6

Section 5

Section 3

Section 4

Section 2

Figure 5. Representative composite and single-stained images of the multiplex IHC panel in 6 sections in Patient 5. Scale bar: 80 μ m.

Patient 6

Section 1

CD8 CD4 CD20 CD68

CD8

CD4

CD20

CD68

Section 2

CD8 CD4 CD20 CD68

CD8

CD4

CD20

CD68

Section 3

CD8 CD4 CD20 CD68

CD8

CD4

CD20

CD68

Section 4

CD8 CD4 CD20 CD68

CD8

CD4

CD20

CD68

Section 5

CD8 CD4 CD20 CD68

CD8

CD4

CD20

CD68

Section 6

CD8 CD4 CD20 CD68

CD8

CD4

CD20

CD68

Figure 6. Representative composite and single-stained images of the multiplex IHC panel in 6 sections in Patient 6. Scale bar: 80 μ m.

Comment: “4. Assessment of high/low density of subtypes (e.g., CD68+STING+) will be skewed towards the tumors with higher number of imaged fields. Approximately 6,500 fields were used from 80 specimens in the machine learning model. Was each field used as a separate sample or were all fields from a single tumor used as a cluster? How was this defined in the model?”

Reply: Thank you for your comment. First, the density of cells in each ROI was calculated via the normalization of the total cell counts by the total area (cell/mm²). Thus, the densities of immune cell subtypes will not be affected by the number of imaged fields. We provided a cell density calculation demo (code availability) to clarify the analysis process.

Second, all fields from a single tumor were used as a cluster. We added the description in the methods section.

Comment: “5. Are the training and validation cohort sizes large enough (considering 13 immune markers were used) for a meaningful comparison of outcomes? Please provide references if they are any.”

Reply: We appreciate your suggestion. The final prediction model included 4 immune cell phenotypes and 8 immune markers. We have provided several studies with equivalent cohort sizes or even smaller.⁶⁻⁹ Additionally, we stated the following in the limitation section: “Further studies in large cohorts, ideally from prospective clinical trials, are required to confirm our findings.”

Comment: “6. Not sure how commonly the protocol for IHC staining of 13 markers in one tissue section is used in practice. Are there any artefacts from the previous staining on the new ones? I’m assuming the last few staining will have significant artefacts considering some antibodies were used at a very high concentration.”

Reply: We apologize for this confusion. Five consecutive sections (4 µm thick) were cut from FFPE blocks. One section was used for hematoxylin and eosin (H&E) staining. Four FFPE tumor slides (4 µm thick) underwent multiplex immunohistochemistry. For each panel, we stained six markers including CK and DAPI. Thus, the merged picture of four panels is presented in Figure 1c-f. The panels are as follows:

Panel 1: CD8, PD-1, TIM-3, LAG-3, CK, DAPI

Panel 2: CD4, FoxP3, PD-L1, CTLA-4, CK, DAPI

Panel 3: CD68, CD163, HLA-DR, STING, CK, DAPI

Panel 4: CD20, CD66b, CK, DAPI

The protocol for multiplex IHC staining of six markers in one tissue is commonly used in practice. We also cited references in the methods section.¹⁰⁻¹²

Comment: “7. The authors adopted 20µm radius to calculate the effective density as an additional measurement. And, it is said that it was pre-selected and consistent with the previous reports. Please provide more detailed information and references in the methods.”

Reply: We appreciate your suggestion. The radius of 20 µm is consistent with previous reports. As you requested, we cited references in the manuscript.¹¹⁻¹⁴ We also presented the previously analyzed correlation of effective density. Pearson’s correlation coefficients are shown between different radii. The Pearson’s correlation coefficients between 10 and 20 µm and between 20 and 30 µm were nearly 1 (min-max: 0.901–0.998, 0.989–0.999, respectively), indicating a high degree of consistency. However, the correlation between 10 and 30 µm was relatively lower (min-max: 0.846–0.993). Thus, we adopted a 20-µm radius to calculate the effective density, which was also consistent with previous reports.

Figure 7. The correlation of effective density between different radii (Pearson’s chi-squared test)

Comment: “8. What is the dominant predictive biomarker value of the density versus spatial organization for OS and response to immunotherapy?”

Reply: We appreciate your question. First, we quantified the contribution of each marker through feature importance by scikit-learn package (Supplementary Table S22-23). We outputted the feature importance and the average value of each parameter to present its contribution. As shown in Figure 5a, the effective score of CD8⁺PD-1⁺LAG-3⁻ had higher feature importance than the density of CD68⁺STING⁺, CD4⁺FoxP3⁻PD-L1⁺, or CD8⁺PD-1⁻LAG-3⁻ in the ETC, GBC, and ABC machine learning models. As presented in Figure 5b, the effective score of CD8⁺PD-1⁺LAG-3⁻ had a higher feature importance than the other three immune cell types, EBV, MMR, and PD-L1 CPS. Thus, the dominant predictive marker is spatial organization

for response to immunotherapy.

Second, we presented the prognostic biomarkers in predicting OS in Figures 2d and 3d including density and spatial biomarkers. Compared with each single biomarker, the final TIIC score is of great prognostic value (Figure 5c-d).

Comment: “9. Page 4, in the results section, the authors state that 80 patients were enrolled in this study between July 2014 and December 2019. On the other hand, in the methods part, it is described that GC tissues included 85 pre-treatment samples from 80 subjects with histologically confirmed gastric adenocarcinoma collected from March 2018 to December 2020.”

Reply: Thank you for pointing out this problem. The date described in the results section was the diagnosis date of patients. The date described in the methods part is the date of samples collected in the cohort. We revised the date in the methods section to be the same as the diagnosis date in the results section.

Comment: “10. Page 9, line184, the authors describe M1 macrophages as CD68+CD163+HLA-DR-, and M2 macrophages as CD68+HLA-DR+CD163-. However, some other reports describe M2 macrophages as CD163+cells. Please address this discrepancy.”

Reply: We appreciate your question. CD163 is preferentially expressed by monocytes and macrophages. However, recent studies using immunohistochemistry have reported that some dendritic cells also express CD163.^{15,16} The limitation of using a single marker to describe a cell phenotype is the higher false-positive rate. Thus, in this study, we used CD68 as a macrophage marker, HLA-DR as an M1 marker, and CD163 as an M2 marker, which were consistent with a previous study.¹⁷ Thus, we combined these markers to select M1 and M2 phenotypes, which is also consistent with a previous study.¹²

Comment: “11. It would be useful to describe why the stage I/II/III patients enrolled in this study were not eligible for other treatments such as surgery.”

Reply: We apologize for this confusion. Patients with stage I/II/III at diagnosis received surgery according to ESMO/NCCN treatment guidelines.⁵ We collected these patients' samples from surgery (the nearest sample before receiving immunotherapy). These patients received anti-PD1/PD-L1 immunotherapy after tumor recurrence.

We thank you for all the constructive suggestions. We hope you will find that the manuscript has been significantly revised and the concerns have been thoroughly

addressed. Thank you again for your time and effort.

References

1. Pignon JC, Jegede O, Shukla SA, Braun DA, Horak CE, Wind-Rotolo M, Ishii Y, Catalano PJ, Grosha J, Flaifel A, Novak JS, Mahoney KM, Freeman GJ, Sharpe AH, Hodi FS, Motzer RJ, Choueiri TK, Wu CJ, Atkins MB, McDermott DF, Signoretti S. irRECIST for the Evaluation of Candidate Biomarkers of Response to Nivolumab in Metastatic Clear Cell Renal Cell Carcinoma: Analysis of a Phase II Prospective Clinical Trial. *Clin Cancer Res*. 2019 Apr 1;25(7):2174-2184.
2. Gide TN, Quek C, Menzies AM, Tasker AT, Shang P, Holst J, Madore J, Lim SY, Velickovic R, Wongchenko M, Yan Y, Lo S, Carlino MS, Guminski A, Saw RPM, Pang A, McGuire HM, Palendira U, Thompson JF, Rizos H, Silva IPD, Batten M, Scolyer RA, Long GV, Wilmott JS. Distinct Immune Cell Populations Define Response to Anti-PD-1 Monotherapy and Anti-PD-1/Anti-CTLA-4 Combined Therapy. *Cancer Cell*. 2019 Feb 11;35(2):238-255.e6.
3. Nakamura Y, Kawazoe A, Lordick F, Janjigian YY, Shitara K. Biomarker-targeted therapies for advanced-stage gastric and gastro-oesophageal junction cancers: an emerging paradigm. *Nat Rev Clin Oncol*. 2021 Aug;18(8):473-487.
4. Kim R, An M, Lee H, Mehta A, Heo YJ, Kim KM, Lee SY, Moon J, Kim ST, Min BH, Kim TJ, Rha SY, Kang WK, Park WY, Klempner SJ, Lee J. Early Tumor-Immune Microenvironmental Remodeling and Response to First-Line Fluoropyrimidine and Platinum Chemotherapy in Advanced Gastric Cancer. *Cancer Discov*. 2022 Apr 1;12(4):984-1001.
5. Smyth EC, Nilsson M, Grabsch HI, van Grieken NC, Lordick F. Gastric cancer. *Lancet*. 2020 Aug 29;396(10251):635-648.
6. Gonçalves-Ribeiro S, Sanz-Pamplona R, Vidal A, Sanjuan X, Guillen Díaz-Maroto N, Soriano A, Guardiola J, Albert N, Martínez-Villacampa M, López I, Santos C, Serra-Musach J, Salazar R, Capellà G, Villanueva A, Molleví DG. Prediction of pathological response to neoadjuvant treatment in rectal cancer with a two-protein immunohistochemical score derived from stromal gene-profiling. *Ann Oncol*. 2017 Sep 1;28(9):2160-2168.
7. Han J, Duan J, Bai H, Wang Y, Wan R, Wang X, Chen S, Tian Y, Wang D, Fei K, Yao Z, Wang S, Lu Z, Wang Z, Wang J. TCR Repertoire Diversity of Peripheral PD-1+CD8+ T Cells Predicts Clinical Outcomes after Immunotherapy in Patients with Non-Small Cell Lung Cancer. *Cancer Immunol Res*. 2020 Jan;8(1):146-154.
8. Ohue Y, Kurose K, Karasaki T, Isobe M, Yamaoka T, Futami J, Irei I, Masuda T, Fukuda M, Kinoshita A, Matsushita H, Shimizu K, Nakata M, Hattori N, Yamaguchi H, Fukuda M, Nozawa R, Kakimi K, Oka M. Serum Antibody Against NY-ESO-1 and XAGE1 Antigens Potentially Predicts Clinical Responses to Anti-Programmed Cell Death-1 Therapy in NSCLC. *J Thorac Oncol*. 2019 Dec;14(12):2071-2083.
9. Malczewska A, Frampton AE, Mato Prado M, Ameri S, Dabrowska AF, Zagorac S, Clift AK, Kos-Kudła B, Faiz O, Stebbing J, Castellano L, Frilling A. Circulating MicroRNAs in Small-bowel Neuroendocrine Tumors: A Potential Tool for Diagnosis and Assessment of Effectiveness of Surgical Resection. *Ann Surg*. 2021 Jul 1;274(1):e1-e9.
10. Berry S, Giraldo NA, Green BF, Cottrell TR, Stein JE, Engle EL, Xu H, Ogurtsova A, Roberts C, Wang D, Nguyen P, Zhu Q, Soto-Diaz S, Loyola J, Sander IB, Wong PF, Jessel S, Doyle J, Signer D,

Wilton R, Roskes JS, Eminizer M, Park S, Sunshine JC, Jaffee EM, Baras A, De Marzo AM, Topalian SL, Kluger H, Cope L, Lipson EJ, Danilova L, Anders RA, Rimm DL, Pardoll DM, Szalay AS, Taube JM. Analysis of multispectral imaging with the AstroPath platform informs efficacy of PD-1 blockade. *Science*. 2021 Jun 11;372(6547):eaba2609.

11. Huang YK, Wang M, Sun Y, Di Costanzo N, Mitchell C, Achuthan A, Hamilton JA, Busuttill RA, Boussioutas A. Macrophage spatial heterogeneity in gastric cancer defined by multiplex immunohistochemistry. *Nat Commun*. 2019 Sep 2;10(1):3928.

12. Carstens JL, Correa de Sampaio P, Yang D, Barua S, Wang H, Rao A, Allison JP, LeBleu VS, Kalluri R. Spatial computation of intratumoral T cells correlates with survival of patients with pancreatic cancer. *Nat Commun*. 2017 Apr 27;8:15095.

13. Väyrynen JP, Haruki K, Väyrynen SA, Lau MC, Dias Costa A, Borowsky J, Zhao M, Ugai T, Kishikawa J, Akimoto N, Zhong R, Shi S, Chang TW, Fujiyoshi K, Arima K, Twombly TS, Da Silva A, Song M, Wu K, Zhang X, Chan AT, Nishihara R, Fuchs CS, Meyerhardt JA, Giannakis M, Ogino S, Nowak JA. Prognostic significance of myeloid immune cells and their spatial distribution in the colorectal cancer microenvironment. *J Immunother Cancer*. 2021 Apr;9(4):e002297.

14. Väyrynen SA, Zhang J, Yuan C, Väyrynen JP, Dias Costa A, Williams H, Morales-Oyarvide V, Lau MC, Rubinson DA, Dunne RF, Kozak MM, Wang W, Agostini-Vulaj D, Drage MG, Brais L, Reilly E, Rahma O, Clancy T, Wang J, Linehan DC, Aguirre AJ, Fuchs CS, Coussens LM, Chang DT, Koong AC, Hezel AF, Ogino S, Nowak JA, Wolpin BM. Composition, Spatial Characteristics, and Prognostic Significance of Myeloid Cell Infiltration in Pancreatic Cancer. *Clin Cancer Res*. 2021 Feb 15;27(4):1069-1081.

15. Bourdely P, Anselmi G, Vaivode K, Ramos RN, Missolo-Koussou Y, Hidalgo S, Tosselo J, Nuñez N, Richer W, Vincent-Salomon A, Saxena A, Wood K, Lladser A, Piaggio E, Helft J, Guernonprez P. Transcriptional and Functional Analysis of CD1c+ Human Dendritic Cells Identifies a CD163+ Subset Priming CD8+CD103+ T Cells. *Immunity*. 2020 Aug 18;53(2):335-352.e8.

16. Maniecki MB, Møller HJ, Moestrup SK, Møller BK. CD163 positive subsets of blood dendritic cells: the scavenging macrophage receptors CD163 and CD91 are coexpressed on human dendritic cells and monocytes. *Immunobiology*. 2006;211(6-8):407-17.

17. Shaikh S, Noshirwani A, West N, Perry S, Jayne D. Can macrophages within the microenvironment of locally invasive rectal cancers predict response to radiotherapy? *Lancet*. 2015 Feb 26;385 Suppl 1:S87.

Reviewer #3 (Remarks to the Author):

The study employs a newly developed multiplex immunohistochemistry approach to provide insight in the location of immune populations in gastric cancer to explore their potential to predict response to PDL1/PD1 immunotherapy. Based on their findings, a tumor-infiltrating immune cell signature was compiled out of the density of CD4⁺FoxP3⁻PD-L1⁺ T cells, CD8⁺PD-1⁻LAG3⁻ T cells, and CD68⁺STING⁺ macrophages, and the effective score of CD8⁺PD-1⁺LAG3⁻ T cells. Analysis revealed that this signature had predictive value for response to PDL1/PD1 therapy and a priori anti-cancer immune responses. Finally, the immune composition was studied before and after PD1/PDL1 therapy to show that CD8⁺PD1⁻ cells increased, but STING⁺ macrophages and PD1/PDL1⁺ T cells decreased after therapy. This manuscript currently remains very inaccessible in its presentation.

Comment: “1. The immunohistochemistry data is hard to read, not just because of poor resolution, but also because no effort has been made to point out the relevant cells in the pictures of Fig 1c-f and 5c. It would perhaps also be helpful to use color coding to re-label double or triple positive cells for clarity.”

Reply: We appreciate your suggestion. We pointed out the color-coded double- or triple-positive cells in Figure 2b and Supplementary Figure 2a. As it is hard to point out single cells in Figures 1c-f, we only presented representative composite and single-stained images in four panels, which were consistent with previous studies.¹⁻⁵ The resolution in Figures 1c-f is 300 bpi, which meets the standard for journal publication. Additionally, the reason why the IHC data may be hard to read may be because of the file type; we recommend downloading the original picture.

Comment: “2. The heatmap presentation in Fig 2c and 3c is only directly readable for association with the combined positive score (CPS) and not for any of the other tumor indicators. CPS is also not clearly defined.”

Reply: We appreciate your suggestion. First, we added the detailed generation method of the CPS. The CPS was defined as the number of PD-L1-positive tumor cells (partial or complete membrane staining), lymphocytes, and macrophages (membrane staining or intracellular staining, or both) divided by the total number of viable tumor cells multiplied by 100, which is the same definition as the Checkmate-649 clinical trial.⁶ We added this in the methods section. Second, we presented the heatmap mainly according to CPS in the main figure. As you requested, we added the heatmaps grouped by other tumor indicators (Lauren classification, stage, HER2, EBV, MMR) in the Supplementary Figures S3a-e and S5a-e.

Comment: “3. The presence of a large number of associations of immune cells to other tumor parameters throughout the text actually diffuse the key message of the paper, given that many of them do not directly relate to the prognostic value for PD1/PDL1 therapy. These statements should be placed in this context or otherwise removed.”

Reply: We appreciate your great suggestion. As you requested, we placed the statements in the context of predicting efficacy of immunotherapy. The associations of immune cells and GC molecular parameters will provide guidance to further immunotherapy strategies.

In this study, our primary aim was to evaluate the density and spatial patterns of TIICs in the context of anti-PD-1/PD-L1 treatment for a better understanding of the determinants of response to immunotherapy in GC. We showed the density and spatial patterns of tumor-infiltrating immune cells, and their variation depending on the GC molecular feature. These results will allow us to obtain a better understanding of the whole picture of the gastric cancer microenvironment. Eventually, from all these above parameters, we selected the key TIIC signature.

Comment: “4. The proximity of immune cells to the tumor cells was studied to investigate whether immune cells in close range to tumor cells would be more beneficial or more detrimental for the patient. Impact is observed for macrophages at close range on patient survival and for PD1⁺ CD8 T cells for treatment response to PD1/PDL1 therapy. This latter finding appears important for the key message of the paper but yet only the data on macrophages has been displayed in main figure 3.”

Reply: We appreciate your comment. We presented the association of cancer cell-adjacent macrophages with survival in Figure 3d because the survival difference of macrophages was significant. We also presented other immune cells, including PD1⁺CD8 T cells, with survival in Figure S6c (other markers' survival analyses were not significant). Additionally, we evaluated the prognostic value of the effect score of CD8⁺PD-1⁺LAG-3⁻ immune cells. However, the survival difference based on CD8⁺PD-1⁺LAG-3⁻ immune cells alone is not significant in irPFS or irOS (Supplementary Figure 7c).

Comment: “5. The definition of the tumor-infiltrating immune cell signature actually obscures how the parameters associate with predictive value for PD1/PDL1 therapy. This omission should be clarified.”

Reply: We appreciate your comment. First, we found that the density of CD4⁺FoxP3⁻PD-L1⁺ T cells and the effective score of CD8⁺PD-1⁺LAG-3⁻ T cells were closely associated with a positive response to anti-PD-1/PD-L1 therapy; conversely,

CD8⁺PD-1⁻LAG-3⁻ T cells and CD68⁺STING⁺ macrophages were closely associated with a negative response to anti-PD-1/PD-L1 therapy (Supplementary Table S19). Then, we used the density of CD4⁺FoxP3⁺PD-L1⁺ T cells, CD8⁺PD-1⁻LAG3⁻ T cells, and CD68⁺STING⁺ macrophages, and the effective score of CD8⁺PD-1⁺LAG3⁻ T cells to define a TIIC signature. We used four types of machine learning models and calculated the area under the curve (AUC) of the training and validation cohorts including extra tree classifier (ETC), gradient boosting classifier (GBC), AdaBoost classifier (ABC), and multilayer perceptron (MLP). In the validation cohort, the average AUCs of the four algorithms were 0.80, 0.85, 0.77, and 0.75, respectively. The corresponding 95% confidence intervals were narrow, suggesting that the TIIC signature can indeed be used to predict response to immunotherapy (Supplementary Table S20).

We would like to clarify the details of the construction of the TIIC signature in the following aspects:

First, we aimed to illustrate the prediction ability of TIIC signatures in immunotherapy. Rather than developing a specific prediction model, we constructed machine learning models with random three-fold cross-validation to demonstrate the prediction ability of the TIIC signature. This whole process was repeated 5000 times. Since the building process of each model included random three-fold cross-validation, there are subtle differences in the parameters of each model. Based on the mean AUC value of 5000 prediction results and the corresponding 95% confidence interval, we found that the TIIC signature demonstrates a good performance in predicting the clinical outcome of immunotherapy.

Second, to investigate which marker gives the best predictive power, we analyzed feature importance to quantify the contribution of each indicator in ETC, ABC, and GBC models. The indicators that contributed more to the prediction model had a higher feature importance, and vice versa. We present the average value of feature importance and 95% confidence interval for each indicator in the manuscript (Supplementary Table S22-23). The spatial feature (the effective score of CD8⁺PD-1⁺LAG3⁻ T cells) had the highest feature importance.

Third, we submitted the original code of four machine learning models and provided a demo. It is common practice to demonstrate these machine models by submitting original code, as ensemble classifiers and multilayer perceptron are difficult to describe due to their large number of parameters. We updated the code in revision.

In conclusion, through all the above efforts, we clarified the definition of the tumor-infiltrating immune cell (TIIC) signature. We provided the construction procedure of four machine learning models. We could conclude that the TIIC signature is associated with treatment response in PD-1/PD-L1 therapy. We hope we have clarified the definition of the tumor-infiltrating immune cell signature and how the parameters

associate with predictive value for PD1/PDL1 therapy through all these efforts.

We thank you for all the constructive comments. We hope you will find that the manuscript has been significantly revised with your help, and the concerns have been thoroughly addressed. Thank you again for your time and effort.

References

1. Berry S, Giraldo NA, Green BF, Cottrell TR, Stein JE, Engle EL, Xu H, Ogurtsova A, Roberts C, Wang D, Nguyen P, Zhu Q, Soto-Diaz S, Loyola J, Sander IB, Wong PF, Jessel S, Doyle J, Signer D, Wilton R, Roskes JS, Eminizer M, Park S, Sunshine JC, Jaffee EM, Baras A, De Marzo AM, Topalian SL, Kluger H, Cope L, Lipson EJ, Danilova L, Anders RA, Rimm DL, Pardoll DM, Szalay AS, Taube JM. Analysis of multispectral imaging with the AstroPath platform informs efficacy of PD-1 blockade. *Science*. 2021 Jun 11;372(6547):eaba2609.
2. Huang YK, Wang M, Sun Y, Di Costanzo N, Mitchell C, Achuthan A, Hamilton JA, Busuttill RA, Boussioutas A. Macrophage spatial heterogeneity in gastric cancer defined by multiplex immunohistochemistry. *Nat Commun*. 2019 Sep 2;10(1):3928.
3. Carstens JL, Correa de Sampaio P, Yang D, Barua S, Wang H, Rao A, Allison JP, LeBleu VS, Kalluri R. Spatial computation of intratumoral T cells correlates with survival of patients with pancreatic cancer. *Nat Commun*. 2017 Apr 27;8:15095.
4. Väyrynen JP, Haruki K, Väyrynen SA, Lau MC, Dias Costa A, Borowsky J, Zhao M, Ugai T, Kishikawa J, Akimoto N, Zhong R, Shi S, Chang TW, Fujiyoshi K, Arima K, Twombly TS, Da Silva A, Song M, Wu K, Zhang X, Chan AT, Nishihara R, Fuchs CS, Meyerhardt JA, Giannakis M, Ogino S, Nowak JA. Prognostic significance of myeloid immune cells and their spatial distribution in the colorectal cancer microenvironment. *J Immunother Cancer*. 2021 Apr;9(4):e002297.
5. Väyrynen SA, Zhang J, Yuan C, Väyrynen JP, Dias Costa A, Williams H, Morales-Oyarvide V, Lau MC, Rubinson DA, Dunne RF, Kozak MM, Wang W, Agostini-Vulaj D, Drage MG, Brais L, Reilly E, Rahma O, Clancy T, Wang J, Linehan DC, Aguirre AJ, Fuchs CS, Coussens LM, Chang DT, Koong AC, Hezel AF, Ogino S, Nowak JA, Wolpin BM. Composition, Spatial Characteristics, and Prognostic Significance of Myeloid Cell Infiltration in Pancreatic Cancer. *Clin Cancer Res*. 2021 Feb 15;27(4):1069-1081.
6. Jangjigian Yelena Y., Shitara Kohei., Moehler Markus., Garrido Marcelo., Salman Pamela., Shen Lin., Wyrwicz Lucjan., Yamaguchi Kensei., Skoczylas Tomasz., Campos Bragagnoli Arinilda., Liu Tianshu., Schenker Michael., Yanez Patricio., Tehfe Mustapha., Kowalyszyn Ruben., Karamouzis Michalis V., Bruges Ricardo., Zander Thomas., Pazo-Cid Roberto., Hitre Erika., Feeney Kynan., Cleary James M., Poulart Valerie., Cullen Dana., Lei Ming., Xiao Hong., Kondo Kaoru., Li Mingshun., Ajani Jaffer A.(2021). First-line nivolumab plus chemotherapy versus chemotherapy alone for advanced gastric, gastro-oesophageal junction, and oesophageal adenocarcinoma (CheckMate 649): a randomised, open-label, phase 3 trial. *Lancet*, 398(10294), 27-40.

Reviewer #4 (Remarks to the Author):

MAJOR COMMENTS

Comment: “1. Line 229: The authors applied four different algorithms to predict response of gastric cancer to immunotherapy and report the maximum and minimum area under the receiver operating characteristic curve. However, there are only 15 subjects in the validation set because among the 16 in the validation set, one was missing ORR. Therefore, the 95% CIs should be reported for each algorithm rather than the point estimates of the AUC for the 4 algorithms. Also, lines 230-232 describe which methods perform better. It would be better if for each of the four methods the AUC and 95% CI were reported rather than simply a min-max range.”

Reply: We appreciate your helpful suggestion. As you requested, we added the 95% CIs for each of the methods including AUC and 95% CI in Figure 4b and Supplementary Tables 20-21. We revised the description in the results section. Additionally, we provided several studies with equivalent or smaller validation cohort sizes.¹⁻³ We also stated in the limitation section that further studies in large cohorts, ideally from prospective clinical trials, are required to confirm our findings.

Comment: “2. Lines 213-215: The authors wanted to ensure balance between the training and validation sets. One patient was excluded from the validation set (see lines 476-477). It would seem more appropriate if Table 2 compared the 15 patients actually used in validation rather than including one patient that was subsequently omitted.”

Reply: We appreciate your suggestion. As you requested, we showed the baseline characteristics in 15 patients in the validation sets in Table 2. Please see the revised Table 2.

Comment: “3. Lines 236-239: The authors should report the AUC for tumor mutation burden, PD-L1 expression, microsatellite instability, and Epstein-Barr virus infection status and a multivariable model for these given they have been proposed to identify susceptibility to PD-1/PD-L1 inhibitors. What about HER2 (omitted from this section)? That is, the authors should demonstrate their biomarkers offer some advantage over current ones. It does not seem to be a fair comparison to combine their biomarkers but analyze the others univariately.”

Reply: We appreciate your suggestion. We presented the AUC for the PD-L1 CPS in a supplementary figure. As microsatellite instability (MSI), Epstein-Barr virus (EBV) infection status, and HER2 expression were expressed as binary variables, we

analyzed the treatment response based on EBV status (positive vs. negative), MMR status (pMMR vs. dMMR), and HER2 expression (positive vs. negative) in univariate and multivariable logistic regression models (Supplementary Table S27). As we can see from univariate and multivariable models, EBV-positive status and dMMR tended to be associated with better response. The association of HER2 expression with treatment response was not consistent between univariate and multivariable models. Therefore, taken together, our data suggest that the TIIC signature has greater power for patient stratification.

Tumor mutational burden (TMB) reflects cancer mutation quantity. Nevertheless, TMB is an imperfect response biomarker.⁴ A linear relationship between TMB and immune checkpoint inhibitor responsiveness has been described.⁵ However, there is currently no consensus in the definition of TMB cut-offs for patient stratification. In addition, there is limited consensus regarding the panel design and weights of each gene. Plus, only a small fraction of non-synonymous mutations will result in neo-antigens that are recognized by T-cells. The clonality of these neo-antigens and the specific tumor molecular signatures contribute to the ability to generate a unique and effective anti-tumor response. TMB data in our cohort is limited, and unfortunately, we could not report the AUC for TMB.

In addition, we added PD-L1 CPS score, MSI, and EBV status with TIIC in the machine learning models to predict treatment response as a combined TIIC signature. The AUC of the combined TIIC signature improved in the three models except for the MLP model – please see the revision in Figure 4b-c, and Supplementary Table S21. We also quantified the contribution of each marker in the prediction through feature importance using the scikit-learn package (Supplementary Table S22-23). We outputted the feature importance and the average value of each parameter to present its contribution. As shown in Figure 5a, the effective score of CD8⁺PD-1⁺LAG-3⁻ had a higher feature importance than the density of CD68⁺STING⁺, CD4⁺FoxP3⁻PD-L1⁺, or CD8⁺PD-1⁻LAG-3⁻ in the ETC, GBC, and ABC machine learning models. As presented in Figure 5b, the effective score of CD8⁺PD-1⁺LAG-3⁻ had a higher feature importance than the other three immune cell types, EBV status, MMR, and PD-L1 CPS. Thus, the dominant predictive marker is spatial organization for response to immunotherapy.

Comment: “4. Throughout: The expression of 16 proteins was assessed along with the density and spatial location of immune cells. Tumor core (TC) [N=4477], invasion margin (IM) [N=993], and peritumoral normal areas (N) [N=1018] were reported as regions of interest. Despite this, the authors reported p-values with seemingly no adjustment for multiple hypothesis tests. The authors need to use some adjustment for multiple comparisons or alternatively report the false discovery rates instead of the raw p-values. Also, throughout the manuscript and figures, “ns” should be replaced with the appropriate p-value or FDR.”

Reply: Thank you for your helpful suggestion. First, all fields from a single tumor were used as a cluster. We added this statement in the methods section. Second, the densities were continuous and non-normally distributed and did not meet the assumptions of parametric analyses. Thus, we used the Kruskal–Wallis test with the Dunn multiple comparison test to compare density differences among TC, IM, and N for multiple comparisons in Figure 2a-b and Supplementary Figure 1b.⁶ We revised all relevant results and methods. Third, we replaced “ns” with *P*-value as you requested throughout the manuscript and figures.

Comment: “5. Lines 242-257: When deriving the high- vs low-score groups, it is unclear whether Cox PH models were fit to a training dataset or to the entire cohort of patients. Based on Figures 2/3/4, it seems the latter was used. If indeed the entire cohort was used, the scores need to be re-estimated by fitting models only to the training data and then applying those score to the validation data. Only the validation results are of intrinsic interest. Additionally, there is no utility in presenting results for the combined training+validation sets.”

Reply: We appreciate your suggestion. When deriving the high- and low-score groups, Cox PH models were fit to a training dataset and then those scores were applied to the validation data. Figures 2 and 3 mainly focused on the overall tumor immune features in 80 patients. We apologize for this confusion.

We revised Figure 5b to only present validation results as you suggested. The revised Figures 4 and 5 mainly exhibit the prediction of TIIC in patients receiving immunotherapy.

MINOR COMMENTS

Comment: “6. Figure 2: It would be helpful if Tumor core (TC), invasion margin (IM), and peritumoral normal areas (N) were added to the caption.”

Reply: Thank you for reminding us of the omission. We added “Tumor core (TC), invasion margin (IM), and peritumoral normal areas (N)” in the caption of Figure 2.

Comment: “7. Figure 4: The ROC for the training data is not of intrinsic interest.”

Reply: We appreciate your suggestion. We deleted the ROC of training data and only present validation results in Figures 4 and 5 as per your suggestion.

Comment: “8. The title, “Predicting response to immunotherapy in gastric cancer...,” may not be entirely reflective of the manuscript’s content given 20 of the 80 gastric

cancer patients did not receive immunotherapy. Survival was assessed in all 80.”

Reply: We appreciate your suggestion. The key message of this study is the multi-dimensional analyses of the tumor immune microenvironment to predict response to immunotherapy. The associations of immune cells with survival allow us to better understand the gastric cancer environment. Then, from all these parameters, we selected the key TIIC signature. Thus, we think the current title may be appropriate and would like to hear your further suggestions.

Comment: “9. According to lines 73-74, five of the 80 patients contributed both pre- and post-treatment samples. However, lines 389-390 state, “GC tissues included 85 pre-treatment samples from 80 subjects.” These statements seem to be conflicting.”

Reply: We apologize for the mistake. We deleted Figure 5 for its limited contribution in the main hypothesis (5 patients pre- and post-treatment samples) as reviewer #1 suggested. Thus, we revised this section accordingly.

Comment: “10. Lines 118-121: The authors tested for whether there was a significant difference between groups but then state, “In general, patients showed similar densities of TIICs according to the Lauren classification, tumor differentiation, and tumor location.” Absence of evidence is not evidence of absence. This needs to be restated that generally, there are few significant differences between Lauren classification, tumor differentiation, and tumor location with respect to densities of TIICs.”

Reply: Thank you very much for your helpful suggestion. We restated that sentence as you suggested.

Comment: “11. Lines 124-125: Why did the authors exclude CD4+, CD4+FoxP3-, and CD68+HLA-DR+CD163- from this sentence?”

Reply: We appreciate your suggestion. According to the revised results in the supplementary tables, we revised the description as follows:

“Overall, the density of total CD8⁺, CD4⁺, CD68⁺, CD20⁺, and CD66b⁺ cells was associated with the disease stage. Additionally, advanced-stage GC (III-IV) samples showed a higher density of exhausted CD8 T cells, Treg cells, and so on.”

Comment: “12. Lines 127-128: Why did the authors exclude CD68+CD163+HLA-DR-STING+ from this statement?”

Reply: We appreciate your suggestion. According to the revised results in the supplementary tables, we revised the description as follows:

“Interestingly, EBV-positive tumors showed higher densities of CD8⁺PD-1⁻LAG-3⁻ T cells than EBV-negative ones.”

Comment: “13. Lines 128-130: Why did the authors exclude CD4⁺, CD4⁺FoxP3⁻, and CD68⁺STING⁺?”

Reply: We appreciate your suggestion. According to the revised results in the supplementary tables, we revised the description as follows:

“Proficient MMR (pMMR) tumors showed a significantly higher abundance of total CD8⁺, CD4⁺, CD68⁺, CD20⁺, and CD66b⁺ cells than dMMR tumors.”

Comment: “14. Lines 130-131: It is unclear how “combined positive score” is derived to then come up with the four groups. Depending on how it was formed, it may or may not make sense to report Supplementary Table S8.”

Reply: We appreciate your suggestion. We added the detailed generation method of the CPS. The CPS was defined as the number of PD-L1-positive tumor cells (partial or complete membrane staining), lymphocytes, and macrophages (membrane staining or intracellular staining, or both) divided by the total number of viable tumor cells multiplied by 100, which is the same definition as that in the Checkmate-649 clinical trial.⁷ We added this in the methods section. We believe it makes sense to report Supplementary Table S8.

Comment: “15. Lines 143-145: The interpretation that the prognostic value was significantly adjusted is not correct. The authors need to reword this sentence to reflect that those tumor-infiltrating T cell subsets were significantly associated with overall survival.”

Reply: Thank you for your suggestion. We revised as follows:

“CD8⁺PD-1⁺LAG-3⁺TIM-3⁺ cells [high vs. low, hazard ratio (HR) 1.98, 95% confidence interval (CI; 1.12–3.50)] and CD68⁺STING⁺ cells [high vs. low, HR 1.83, 95%CI (1.01–3.33)] were significantly associated with OS, as revealed by multivariate Cox analysis (Supplementary Table S9).”

Comment: “16. Lines 181-187: It is not clear how effective scores were assigned and whether they are continuous and meet parametric assumptions in order for Student’s t-test and ANOVA are appropriate.”

Reply: Thank you very much for your comment. We rechecked the effective scores. The effective scores were continuous and non-normally distributed and did not meet the assumptions of parametric analyses. Thus, we used non-parametric analysis (Mann–Whitney U or Kruskal–Wallis test) to compare differences between sub-groups. We updated Supplementary Table S10-17 and revised the results section and methods section.

Comment: “17. Line 404: The authors should clearly articulate what subjects were censored when analyzing immune-related progression-free survival.”

Reply: We appreciate the reviewer’s suggestion. We clarified the description in the methods section as follows:

“Immune-related progression-free survival (irPFS) was defined as the time from initial immunotherapy to the day of disease progression, death, or the end of follow-up, whichever occurred first.”

In this study, 16 patients in our cohort are still being treated and thus were categorized as censored.

Comment: “18. Lines 464-465: The authors should cite relevant references for each method.”

Reply: We appreciate your suggestion and cited relevant references for each method throughout the manuscript.

Comment: “19. Lines 466-467: For each method, the authors should state what hyperparameters were tuned.”

Reply: We appreciate your suggestion. Please see the tuned parameters in Supplementary Table S27.

Comment: “20. Lines 473-475: First, it is unclear what parts of the process were repeated 5000 times. If repeated 3 fold-CV did not also include variable selection, the results will not reflect generalization performance.”

Reply: We appreciate your comment. We revised the following aspects:

First, we performed 5000 repetitions of the whole prediction process. The grouping method of three-fold cross-validation was the same for each time, which can be seen from the code demo of the first submission. Ensemble classifiers that employ decision trees as weak classifiers have randomized establishment and prediction processes, which is one reason for the instability in the results. This instability may overestimate or underestimate the predictive value of the signature. In fact, many previous machine learning-based studies only show the prediction process once, which may be the best performance due to randomization. Therefore, we provided the average AUC of 5000 predictions, which helps to correctly assess the predictive value of the signature.

Second, based on your suggestion on variable selection, we have made major revisions as follows:

- (1) The sample was randomly divided into three groups in three-fold cross-validation for selecting hyperparameters, which is different from that in the first submission.
- (2) The feature importance of each indicator was recorded in 5000 predictions.

Figure 4 and Supplementary Table S20 show that the model based on the three-fold cross-validation with randomized grouping still demonstrates a good performance, which means that the good performance is not a special case by accident, but a generalized performance.

Comment: “21. Lines 485-486: To examine the potential relationship between “each” THIC and GC survival, the authors should fit univariable not multivariable Cox PH models. Also, when fitting Cox PH models the authors should verify the PH assumption and also when fitting multivariable Cox PH models, the authors need to disclose their process for arriving at a final model and then checking model fit.”

Reply: We appreciate the reviewer’s suggestion. We verified the PH assumption previously. We added the following statement in the Methods section:

“The assumption of proportionality of hazards was assessed by a time-varying covariate in the Cox models with a cross-product term of survival time and each THIC. The proportionality of hazards assumptions were generally satisfied for survival ($P > 0.05$).”

Comment: “22. Throughout: Many paragraphs start with something like “Based on the above results.” When in print, those results may not be “above” so these phrases should be re-worded.”

Reply: We appreciate your helpful suggestion. We deleted “Based on the above results” and revised the sentence accordingly.

We thank you for all your constructive comments. We hope you will find that the

manuscript has been significantly revised and the concerns have been thoroughly addressed. Thank you again for your time and effort.

References

1. Gonçalves-Ribeiro S, Sanz-Pamplona R, Vidal A, Sanjuan X, Guillen Díaz-Maroto N, Soriano A, Guardiola J, Albert N, Martínez-Villacampa M, López I, Santos C, Serra-Musach J, Salazar R, Capellà G, Villanueva A, Molleví DG. Prediction of pathological response to neoadjuvant treatment in rectal cancer with a two-protein immunohistochemical score derived from stromal gene-profiling. *Ann Oncol*. 2017 Sep 1;28(9):2160-2168.
2. Han J, Duan J, Bai H, Wang Y, Wan R, Wang X, Chen S, Tian Y, Wang D, Fei K, Yao Z, Wang S, Lu Z, Wang Z, Wang J. TCR Repertoire Diversity of Peripheral PD-1+CD8+ T Cells Predicts Clinical Outcomes after Immunotherapy in Patients with Non-Small Cell Lung Cancer. *Cancer Immunol Res*. 2020 Jan;8(1):146-154.
3. Ohue Y, Kurose K, Karasaki T, Isobe M, Yamaoka T, Futami J, Irei I, Masuda T, Fukuda M, Kinoshita A, Matsushita H, Shimizu K, Nakata M, Hattori N, Yamaguchi H, Fukuda M, Nozawa R, Kakimi K, Oka M. Serum Antibody Against NY-ESO-1 and XAGE1 Antigens Potentially Predicts Clinical Responses to Anti-Programmed Cell Death-1 Therapy in NSCLC. *J Thorac Oncol*. 2019 Dec;14(12):2071-2083.
4. Jardim DL, Goodman A, de Melo Gagliato D, Kurzrock R. The Challenges of Tumor Mutational Burden as an Immunotherapy Biomarker. *Cancer Cell*. 2021 Feb 8;39(2):154-173.
5. Goodman AM, Kato S, Bazhenova L, Patel SP, Frampton GM, Miller V, Stephens PJ, Daniels GA, Kurzrock R. Tumor Mutational Burden as an Independent Predictor of Response to Immunotherapy in Diverse Cancers. *Mol Cancer Ther*. 2017 Nov;16(11):2598-2608.
6. Yue JK, Yuh EL, Korley FK, Winkler EA, Sun X, Puffer RC, Deng H, Choy W, Chandra A, Taylor SR, Ferguson AR, Huie JR, Rabinowitz M, Puccio AM, Mukherjee P, Vassar MJ, Wang KKW, Diaz-Arrastia R, Okonkwo DO, Jain S, Manley GT; TRACK-TBI Investigators. Association between plasma GFAP concentrations and MRI abnormalities in patients with CT-negative traumatic brain injury in the TRACK-TBI cohort: a prospective multicentre study. *Lancet Neurol*. 2019 Oct;18(10):953-961.
7. Janjigian Yelena Y., Shitara Kohei., Moehler Markus., Garrido Marcelo., Salman Pamela., Shen Lin., Wyrwicz Lucjan., Yamaguchi Kensei., Skoczylas Tomasz., Campos Bragagnoli Arinilda., Liu Tianshu., Schenker Michael., Yanez Patricio., Tehfe Mustapha., Kowalyszyn Ruben., Karamouzis Michalis V., Bruges Ricardo., Zander Thomas., Pazo-Cid Roberto., Hitre Erika., Feeney Kynan., Cleary James M., Poulart Valerie., Cullen Dana., Lei Ming., Xiao Hong., Kondo Kaoru., Li Mingshun., Ajani Jaffer A.(2021). First-line nivolumab plus chemotherapy versus chemotherapy alone for advanced gastric, gastro-oesophageal junction, and oesophageal adenocarcinoma (CheckMate 649): a randomised, open-label, phase 3 trial. *Lancet*, 398(10294), 27-40.

REVIEWERS' COMMENTS

Reviewer #1 (Remarks to the Author):

In their revised manuscript the authors have taken considerable effort to try and address the reviewer comments. The current version represents a substantial improvement and my prior comments have now been adjusted. Although the findings are of interest and aligned with themes observed in other tumors there remains significant validation prior to further implementation of predictive IC signatures. It would be of interest to the readership if the authors included the validate ImmunoScore as a comparator against their more complex TIIC signatures. Please see the below reference (and references within) to show how score is done and how it can be incorporated

Antoniotti C, Rossini D, Pietrantonio F, Catteau A, Salvatore L, Lonardi S, Boquet I, Tamberi S, Marmorino F, Moretto R, Ambrosini M, Tamburini E, Tortora G, Passardi A, Bergamo F, Kassambara A, Sbarrato T, Morano F, Ritorto G, Borelli B, Boccaccino A, Conca V, Giordano M, Ugolini C, Fieschi J, Papadopulos A, Massoué C, Aprile G, Antonuzzo L, Gelsomino F, Martinelli E, Pella N, Masi G, Fontanini G, Boni L, Galon J, Cremolini C; GONO Foundation Investigators. Upfront FOLFOXIRI plus bevacizumab with or without atezolizumab in the treatment of patients with metastatic colorectal cancer (AtezoTRIBE): a multicentre, open-label, randomised, controlled, phase 2 trial. *Lancet Oncol.* 2022 May 27;S1470-2045(22)00274-1. doi: 10.1016/S1470-2045(22)00274-1. Epub ahead of print. PMID: 35636444.

Reviewer #2 (Remarks to the Author):

No additional comments

Reviewer #3 (Remarks to the Author):

The authors have provided adequate responses to my questions regarding the accessibility of the manuscript and have implemented the recommendations to improve their paper. I support publication of the work.

Reviewer #4 (Remarks to the Author):

The authors have addressed my previous comments and I have no further concerns.

REVIEWER COMMENTS

Reviewer #1 (Remarks to the Author):

Comment: “In their revised manuscript the authors have taken considerable effort to try and address the reviewer comments. The current version represents a substantial improvement and my prior comments have now been adjusted. Although the findings are of interest and aligned with themes observed in other tumors there remains significant validation prior to further implementation of predictive IC signatures. It would be of interest to the readership if the authors included the validate ImmunoScore as a comparator against their more complex TIIC signatures. Please see the below reference (and references within) to show how score is done and how it can be incorporated.

Antoniotti C, Rossini D, Pietrantonio F, Catteau A, Salvatore L, Lonardi S, Boquet I, Tamberi S, Marmorino F, Moretto R, Ambrosini M, Tamburini E, Tortora G, Passardi A, Bergamo F, Kassambara A, Sbarrato T, Morano F, Ritorto G, Borelli B, Boccaccino A, Conca V, Giordano M, Ugolini C, Fieschi J, Papadopulos A, Massoué C, Aprile G, Antonuzzo L, Gelsomino F, Martinelli E, Pella N, Masi G, Fontanini G, Boni L, Galon J, Cremolini C; GONO Foundation Investigators. Upfront FOLFOXIRI plus bevacizumab with or without atezolizumab in the treatment of patients with metastatic colorectal cancer (AtezoTRIBE): a multicentre, open-label, randomised, controlled, phase 2 trial. *Lancet Oncol.* 2022 May 27:S1470-2045(22)00274-1. doi: 10.1016/S1470-2045(22)00274-1. Epub ahead of print. PMID: 35636444.”

Reply: We appreciate your positive comments. First, we would like to highlight that the tumour microenvironment in gastric and colorectal cancers is largely different. The tumour microenvironment in gastric cancer is notably more complex than that in other gastrointestinal cancers and breast cancer; this complexity has led to the failure of many clinical trials for gastric cancer.

Second, as you suggested, we studied how the ImmunoScore was built in clinical trials (AtezoTRIBE). ImmunoScore IC measures the densities of PD-L1 and CD8 cells as well as the proximity between these cells in a single tissue section. In our study, we collected five consecutive sections from one patient. One section was used for H&E staining and four sections were stained as follows:

Panel 1: CD8, PD-1, TIM-3, LAG-3, CK, DAPI

Panel 2: CD4, FoxP3, PD-L1, CTLA-4, CK, DAPI

Panel 3: CD68, CD163, HLA-DR, STING, CK, DAPI

Panel 4: CD20, CD66b, CK, DAPI

We used the four panels to present the whole picture of the complex tumour

microenvironment in gastric cancer. We tried our best to build the ImmunoScore for our cohort. The density of CD8 and PD-L1 cells was calculated. However, as CD8 and PD-L1 were in different sections in our cohort, we could not evaluate the proximity between these cells because of the limitation of the current technology. Thus, we built a simple version of ‘ImmunoScore’ based only on cell density information.

The area under the curve (AUC) of CD8 (density), PD-L1 cells (density), and CD8 and PD-L1 cells together was 0.53, 0.44, and 0.59, respectively (Figure 1a-c). Furthermore, we built a simple version of the ‘risk score’, characterised by the high density of CD8 and PD-L1 cells, defined as the ‘high’ group, whereas other patients were categorised in the ‘low’ group. As is evident from Figure 1d-e, there were no significant differences in immune-related survival between these two groups. Thus, the simple version of ‘ImmunoScore’ was not sufficiently predictive in gastric cancer patients receiving anti-PD-1/PD-L1 therapy.

Figure 1. Receiver operating characteristic (ROC) curves for the performance of CD8 (a), PD-L1 (b), and combination of CD8 and PD-L1 (c) in gastric cancer (GC) patients subjected to immunotherapy; Kaplan–Meier curves of the irPFS (d) and irOS (e) of the ImmunoScore group. Log-rank (Mantel–Cox) test.

We have cited the Antoniotti et al. article in our manuscript. We are also working on a combination of different sections to build a cross-section spatial microenvironment for a subsequent project. We are sorry that we were unable to achieve the same ‘ImmunoScore’ as in colorectal cancer. However, we believe that our study convincingly demonstrates the fact that the density and spatial distribution of tumour-

infiltrating immune cells, impacted by immunotherapy, can be used to predict the immune responses and prognosis in gastric cancer.

We thank you for the constructive comments and for your time and effort in reviewing our manuscript.

Reviewer #2 (Remarks to the Author):

Comment: “No additional comments.”

Reply: We thank you for your positive comments and for your time and effort in reviewing our manuscript.

Reviewer #3 (Remarks to the Author):

Comment: “The authors have provided adequate responses to my questions regarding the accessibility of the manuscript and have implemented the recommendations to improve their paper. I support publication of the work.”

Reply: We thank you for recommending the acceptance of our manuscript. We are grateful for your time and effort in reviewing our manuscript.

Reviewer #4 (Remarks to the Author):

Comment: “The authors have addressed my previous comments and I have no further concerns.”

Reply: We thank you for accepting our responses to your valuable comments. We express our gratitude for your time and effort in reviewing our manuscript.